# Self-Refining Vision Language Model for Robotic Failure Detection and Reasoning

**Carl Qi**[1], **Xiaojie Wang**[2], **Silong Yong**[3], **Stephen Sheng**[2], **Huitan Mao**[2], **Sriram Srinivasan**[2]
**Manikantan Nambi**[2], **Amy Zhang**[1], **Yesh Dattatreya**[2]
[1] UT Austin    [2] Amazon Robotics    [3] Carnegie Mellon University
`carlq@utexas.edu`

## Abstract

Reasoning about failures is crucial for building reliable and trustworthy robotic systems. Prior approaches either treat failure reasoning as a closed-set classification problem or assume access to ample human annotations. Failures in the real world are typically subtle, combinatorial, and difficult to enumerate, whereas rich reasoning labels are expensive to acquire. We address this problem by introducing ARMOR: Adaptive Round-based Multi-task mOdel for Robotic failure detection and reasoning. We formulate detection and reasoning as a multi-task self-refinement process, where the model iteratively predicts detection outcomes and natural language reasoning conditioned on past outputs. During training, ARMOR learns from heterogeneous supervision - large-scale sparse binary labels and small-scale rich reasoning annotations - optimized via a combination of offline and online imitation learning. At inference time, ARMOR generates multiple refinement trajectories and selects the most confident prediction via a self-certainty metric. Experiments across diverse environments show that ARMOR achieves state-of-the-art performance by improving over the previous approaches by up to 30% on failure detection rate and up to 100% in reasoning measured through LLM fuzzy match score, demonstrating robustness to heterogeneous supervision and open-ended reasoning beyond predefined failure modes. We provide dditional visualizations on our website: `https://sites.google.com/utexas.edu/armor`.

## 1 Introduction

Autonomous robotics has the potential to improve process efficiency and people's well-being, through several applications ranging from warehouse operations (Mitash et al., 2023; Narang et al., 2022) to household and assistive robots (Miller, 2006; Qi et al., 2022; Lin et al., 2022; Black et al., 2024). To achieve true success in these applications, an intelligent robot should not only follow instructions but also reason about task execution and, upon failure, provide explanations that enable human intervention and continuous learning. These capabilities enable downstream applications including generating shaped rewards for reinforcement learning and producing corrective task plans for recovery policies (Duan et al., 2025). While there exist many approaches tackling failure detection, ranging from classical methods (Ye et al., 2019; Garrett et al., 2020) to recent learning-based methods (Duan et al., 2025), they primarily focus on a closed set of failure modes. In reality, failures in robotics are often nuanced, combinatorial, and difficult to enumerate exhaustively. This complexity calls for vision-language models (VLMs) that can leverage broad visual knowledge to reason about novel failure modes in open-ended natural language, rather than task-specific models limited to predefined failure taxonomies. To address these challenges, our work fine-tunes a VLM with custom prediction heads to jointly tackle failure detection and open-ended failure reasoning.

However, training a good failure reasoning model is challenging due to data heterogeneity: while binary failure labels can typically be obtained automatically from system logs, detailed reasoning labels are costly and time-consuming to collect. Recent work AHA (Duan et al., 2025) tackles failure reasoning in robotics by applying supervised instruction fine-tuning (SFT) (Liu et al., 2023a) on a curated dataset of simulated failures with complete reasoning annotations. While effective in this restricted setting, where both binary and reasoning labels are always available for all episodes,

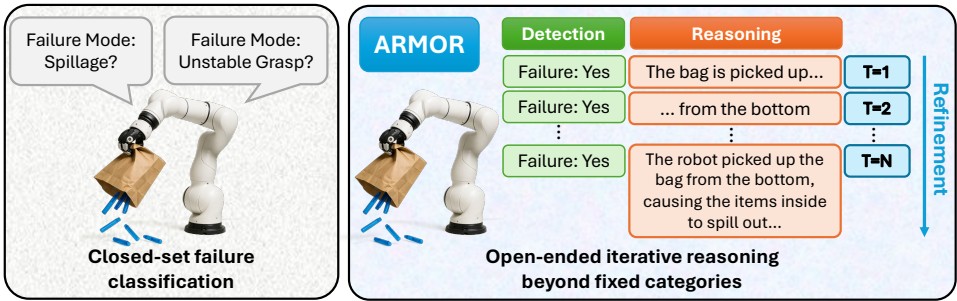

Figure 1: We present ARMOR: adaptive round-based multi-task model for robotic failure detection and reasoning. Prior approaches reduce failure reasoning to closed-set classification of pre-defined modes (e.g., "spillage" or "unstable grasp"). In contrast, ARMOR performs open-ended, iterative refinement by jointly refining its detection and reasoning predictions. This enables accurate detection with nuanced and human-like reasoning that capture real-world failures beyond fixed categories.

this approach suffers from data inefficiency in our heterogeneous data regime. Furthermore, AHA's evaluation relies on hand-crafted regular expressions, making correctness sensitive to ad-hoc format parsing. As shown in our experiments, it also fails to generalize beyond the predefined failure modes it is trained on. In contrast, we introduce a novel fine-tuning algorithm based on multi-round refinement, which scales to heterogeneous supervision and supports open-ended reasoning across diverse and nuanced failure scenarios.

Recent advances in improving foundation models' reasoning capabilities with iterative refinement have shown strong results in domains such as math problems and coding Madaan et al. (2023); Zeng et al. (2023); Qu et al. (2024); Yuan et al. (2024). To our knowledge, we are the first to extend this research frontier to multi-modal failure detection and reasoning with high dimensional video input. In practice, however, naively treating failure detection and reasoning as a single free-form generation task limits classification accuracy and produces inconsistent outputs, as seen in Appendix C. To address this challenge, we present our method, **ARMOR**: **A**daptive **R**ound-based **M**ulti-task m**O**del for **R**obotic failure detection and reasoning. A pictorial illustration of our method compared to prior works is shown in Figure 1. ARMOR treats failure detection and reasoning as a round-based multi-task refinement problem. At each step, the model predicts both a classification outcome and a natural language reasoning, conditions on its previous outputs, and iteratively improves its predictions. Our training algorithm combines expert supervision with online imitation learning (Ross et al., 2011b), leveraging large-scale sparse labels for detection and smaller dense annotations for reasoning, each optimized with task-specific objectives. At inference time, ARMOR generates multiple refinement trajectories and selects the most confident prediction using a metric based on self-certainty (Kang et al., 2025), enabling accurate detection together with open-ended, human-like reasoning.

In summary, the following are the key contributions of this work: 1) We propose a novel approach, ARMOR, that can handle heterogeneous data with open-ended iterative reasoning to perform failure detection and reasoning. 2) To the best of our knowledge, we are the first to leverage iterative reasoning with VLM on multi-modal failure detection in robotics using video feed. 3) Experiments across diverse environments show that ARMOR achieves state-of-the-art performance by improving over the previous approaches by up to 30% on failure detection rate and up to 100% in reasoning measured through LLM fuzzy match score, demonstrating robustness to heterogeneous supervision and open-ended reasoning beyond predefined failure modes.

## 2 RELATED WORK

### 2.1 DETECTING AND REASONING OVER FAILURES IN ROBOTIC MANIPULATION

Failure detection is crucial for monitoring robot execution progress, with research spanning domains from Human-Robot Interaction (HRI) (Ye et al., 2019; Khanna et al., 2023) to Task and Motion Planning (TAMP) (Garrett et al., 2020). Recent advances in LLMs (Hurst et al., 2024; Touvron et al., 2023; Yang et al., 2025) and VLMs (Bai et al., 2025) have enabled the application of foundation

models to visual failure detection and reasoning. Liu et al. (2023b); Zhou et al. (2025) leverage LLMs for failure detection by abstracting visual input into symbolic space or code, but this approach risks losing fine-grained visual cues necessary for detecting subtle failures. Other works rely on off-the-shelf VLMs without any fine-tuning, which limits performance (Skreta et al., 2024; Guo et al., 2024). Finally, prior approaches that fine-tune VLMs for failure detection and reasoning (Duan et al., 2025; Du et al., 2023b) assume a fixed, pre-defined set of failure modes, essentially reducing the task to classification. In contrast, our work focuses on failure detection as a classification problem, and failure reasoning as an open-ended language modeling problem, enabling both accurate predictions and fine-grained, human-like reasoning.

## 2.2 Enhancing the Reasoning Capabilities of Foundation Models

Prior research has explored enhancing the reasoning capabilities of foundation models through various techniques. These include designing prompting strategies for tool interaction (Tihanyi et al., 2025; Wang et al., 2023; Gao et al., 2023), sequential refinement of predictions (Chen et al., 2023; Gou et al., 2024; Zhang et al., 2023), thought verbalization (Nye et al., 2021; Wei et al., 2022; Zhou et al., 2024), and self-critique (Madaan et al., 2023; Shinn et al., 2023) or multi-agent critique (Bai et al., 2022; Du et al., 2023a). While some methods enable self-correction, they often require starting with a strong reasoning model, which is often proprietary (Zhou et al., 2024; Qu et al., 2024). To address this, several works have employed fine-tuning to enhance LLMs' self-improvement capabilities (Zeng et al., 2023; Schick et al., 2023). These approaches leverage preference learning (Rosset et al., 2024), adversarial training (Chen et al., 2024), and reinforcement learning (Yuan et al., 2024; Qu et al., 2024) to learn from self-generated responses. Most closely related to our work is RISE (Qu et al., 2024), as both methods propose a multi-turn self-refinement algorithm and use on-policy data for policy optimization. However, we differ both in setting and algorithm: unlike RISE, we do not assume access to an oracle reward model during training or inference. Instead, we leverage a limited amount of instruction data with a training objective resembling online imitation learning (Ross et al., 2011a), and we rely on the model's internal signals for refinement at inference.

## 2.3 Leveraging Vision-Language Models for Classification

VLMs can be broadly categorized into discriminative models, such as CLIP (Radford et al., 2021) and SigLIP (Zhai et al., 2023), and generative models (Bai et al., 2025; Liu et al., 2023a; Wang et al., 2022), depending on whether the model outputs a representation or generates text or images (Ramesh et al., 2022). Prior work has shown that fine-tuning discriminative VLMs achieves state-of-the-art accuracy on a variety of benchmarks for both in-distribution (ID) and out-of-distribution (OOD) data (Addepalli et al., 2024; Goyal et al., 2023; Kumar et al., 2022). However, discriminative VLMs cannot produce open-ended textual reasoning, making them unsuitable for our task. Recently, Zhang et al. (2024b) demonstrated that generative VLMs can perform on par with, or even surpass, state-of-the-art discriminative VLMs when fine-tuned with classification data, indicating that generative VLMs also encode equally meaningful representations for classification. Building on this insight, our work fine-tunes a generative VLM (Bai et al., 2025) for multitask learning—solving failure detection via classification and failure reasoning via text generation.

## 3 Preliminaries and Problem Setup

### 3.1 Failure Detection and Reasoning as a Multi-Task MDP

Inspired by RISE (Qu et al., 2024), we define a multi-task MDP $\mathcal{M} = (\mathcal{S}, \mathcal{A}, \mathcal{R}, T)$ for our failure detection and reasoning problem. Here, $\mathcal{S}$ denotes the state space, $\mathcal{A}$ the action space, $\mathcal{R} = \{R_{\text{detect}}, R_{\text{reason}}\}$ the reward functions for detection and reasoning, and $T$ the horizon (number of refinement rounds). "Multi-task" in our setting refers to the fact that each action jointly produces two predictions—failure detection and natural language reasoning—rather than separate tasks with independent policies. State $s_t$ consists of input video $x$, prior detection and reasoning predictions $(l^{t-1}, e^{t-1})$, and auxiliary prompt $p$. Action $a_t$ consists of the current multi-task predictions $(l^t, e^t)$,

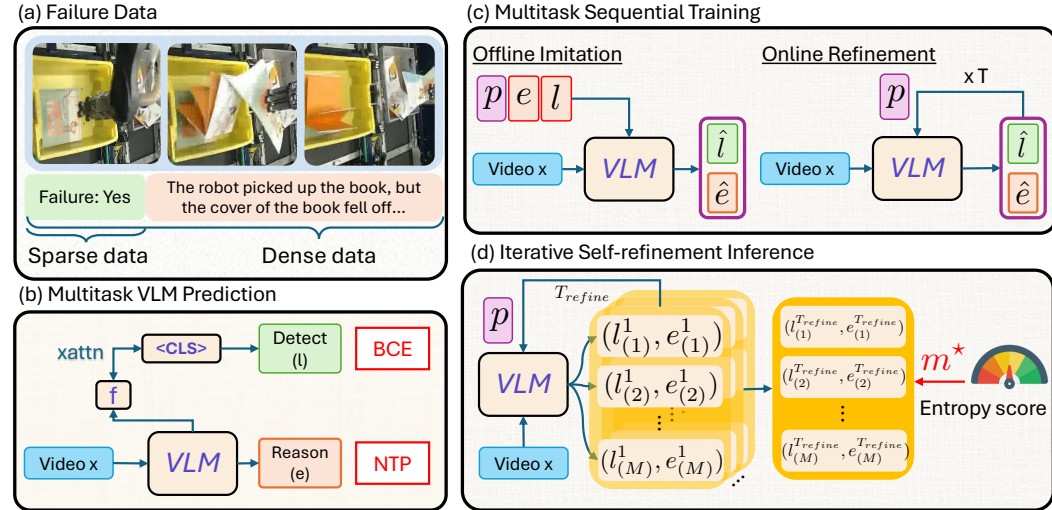

Figure 2: Overview of ARMOR. (a) Our failure data consist of heterogeneous supervision, with large-scale binary detection labels and scarce free-form reasoning labels. (b) A vision-language model (VLM) with multi-task heads jointly predicts detection $l$ via a classification head and reasoning $e$ via a language decoder, trained with binary cross-entropy (BCE) and next-token prediction (NTP) losses. (c) We fine-tune the VLM with both offline imitation and online refinement: the model conditions on dataset labels $(l, e)$ or its prior predictions $(\hat{l}, \hat{e})$ as well as auxiliary prompts $p$ to generate a new round of outputs. In online refinement, this process is repeated $T$ times. The model's predictions (denoted by $\hat{l}$ and $\hat{e}$ in purple rectangles) are supervised with available labels. (d) At inference, ARMOR performs iterative self-refinement, rolling out multiple stochastic trajectories. We select best prediction $m^\star$ from the final predictions with the lowest entropy score.

and the MDP has the following initial condition and transition dynamics:

$$s_0 = [x, \varnothing, \varnothing, \varnothing]$$
$$T(s_{t+1}|s_t, a) = \delta(s_{t+1} = [x, l^t, e^t, p])$$

The *multi-task self-refinement process* is a sequential decision process where the policy refines failure detection and reasoning results for $T$ rounds. At each round $t \in \{0, 1, ..., T-1\}$, it predicts the answer given predictions from the previous round $(l^t, e^t) \sim \pi_\theta \big( \cdot \, \big| \, [x, \, l^{t-1}, \, e^{t-1}, \, p] \big)$. Our goal is to fine-tune a vision-language model (VLM), $\pi_\theta$, to produce human-like reasoning while achieving high failure-detection accuracy after $T$ rounds of refinement. Our reward functions $R_{detect}, R_{reason}$ are defined as the correctness of prediction for a particular task compared to the ground-truth labels $(l, e)$. In practice, we employ different metrics to approximate the correctness when computing performance. We measure detection quality with binary accuracy and reasoning quality with LLM fuzzy match score and Rogue-L score (detailed in Section 5).

## 3.2 Failure Detection and Reasoning with Heterogeneous Supervision

We consider a heterogeneous supervision setting with two datasets: a large-scale *sparse* dataset with binary labels, and a smaller but richer *dense* dataset with detailed human annotations:

$$D_{\text{sparse}} = \{(x_i, l_i)\}, \quad l_i \in \{\text{'success'}, \text{'failure'}\},$$
$$D_{\text{dense}} = \{(x_i, l_i, e_i)\}, \quad e_i \text{ is detailed failure reasoning.}$$

This setting is realistic because binary success/failure labels can often be obtained automatically (e.g., from robot sensors or system logs) and thus scale easily, whereas natural-language reasoning labels require expert annotation and are far more costly. In contrast to prior work AHA (Duan et al., 2025), which assumes dense supervision for both tasks, we operate under a heterogeneous supervision setting, with large-scale binary labels and only a small fraction of reasoning labels, reflecting the practical imbalance faced in real robotic deployments.

## 4 Method

We present ARMOR, which stands for adaptive round-based multi-task refinement of vision-language models (VLMs) for robotic failure detection and reasoning. ARMOR tackles the dual

---

**Algorithm 1** ARMOR Training

---

**Require:** VLM policy $\pi_\theta$; $D_{\text{sparse}}, D_{\text{dense}}$; rounds $T$; task prompts $\{p\}$
    **Phase I: Offline Imitation**
1: *Warm-up:* Train $(l^1, e^1) \sim \pi_\theta(\cdot \mid [x, \varnothing, \varnothing, \varnothing])$ on $D_{\text{sparse}} \cup D_{\text{dense}}$ with BCE for $l$; NTP for $e$.
2: *Expert-conditioned:* Train $(l^1, e^1) \sim \pi_\theta(\cdot \mid [x, l, e, p])$ on $D_{\text{dense}}$ with same losses.
    **Phase II: Online Refinement**
3: **for** mini-batches $(x, l, e)$ from $D_{\text{sparse}} \cup D_{\text{dense}}$ **do**
4:     $(l^0, e^0) \leftarrow (\varnothing, \varnothing)$; $L \leftarrow 0$
5:     **for** $t = 1$ to $T$ **do**
6:         $(l^t, e^t) \sim \pi_\theta(\cdot \mid [x, l^{t-1}, e^{t-1}, p^t])$
7:         $L \mathrel{+}= \text{BCE}(l^t, l) + \text{NTP}(e^t, e) \cdot \mathbf{1}[x \in D_{\text{dense}}]$
8:     **end for**
9:     $\theta \leftarrow \theta - \eta \nabla_\theta L$
10: **end for**

---

challenge of (1) combining classification-style detection with open-ended reasoning, and (2) refining predictions over multiple rounds. To this end, we introduce a model architecture with task-specific prediction heads (Section 4.1). We then cast failure understanding as a *multi-task sequential prediction problem* and train VLMs to condition on prior outputs from each task (Section 4.2). Finally, we design an iterative inference procedure that leverages entropy-based confidence to refine and select final predictions (Section 4.3). An overview of our method is shown in Figure 2.

### 4.1 FAILURE DETECTION AND REASONING VIA MULTI-TASK PREDICTION

Prior work such as AHA (Duan et al., 2025) treats failure detection and reasoning as a single language modeling task, where the model outputs free-form text that consist of both the binary outcome and a short explanation. Evaluation requires hand-crafted regular expressions to extract each answer. This design couples two inherently different objectives and leads to optimization difficulties, since binary classification is optimized through a language modeling loss, and to evaluation fragility, since correctness depends on ad-hoc format parsing rather than explicit comparison.

We instead introduce a multi-task architecture with separate heads for detection and reasoning. Concretely, given a VLM backbone consisting of a vision encoder and language model decoder (e.g., Qwen2.5-VL (Bai et al., 2025)), we follow the established practice (Devlin et al., 2019; Radford et al., 2021) and attach a lightweight *classification head* to the intermediate decoder representation to produce a `[CLS]` token for detection, trained with binary classification. The original *language model head* is retained for reasoning generation, trained with next-token prediction. Both heads share the same vision encoder and language decoder parameters, ensuring consistent representations while allowing task-specific optimization. This separation removes the need for brittle answer extraction, as each prediction head will receive a task-specific input prompt and predicts only task-relevant output. It also enables better supervision under our mixed dense–sparse data regime: dense data provide full labels for both tasks, while sparse data supervise only detection but still encourage reasoning generation as context. The result is a model that optimizes each task appropriately while allowing their outputs to inform one another. Details on model architecture and task-specific prompting are in Appendix A and B.4.

### 4.2 TRAINING VLMS ON MULTI-TASK SEQUENTIAL PREDICTION PATHS

Compared to formulating failure detection and reasoning as a single task of generating free-form natural language, the proposed multi-task heads greatly improve failure classification accuracy. However, failure reasoning quality does not improve and the two heads may output inconsistent results. Two challenges to focus on are improving the consistency of the output of the two heads and the reasoning capability of the model. To address this, ARMOR employs a novel *multi-task self-refinement process* where failure detection and reasoning are iteratively refined. At each round $t \in \{0, 1, ..., T-1\}$, the policy predicts the next round's answer given previous detection and reasoning predictions $(l^t, e^t) \sim \pi_\theta(\cdot \mid [x, l^{t-1}, e^{t-1}, p^t])$. Unlike Qu et al. (2024), we do not assume access to an external reward function. Instead, we directly optimize correctness indicators against

---

**Algorithm 2** ARMOR Inference

---

**Require:** Trained $\pi_\theta$; input $x$; maximum rounds $T_{\text{refine}}$; tolerance $\epsilon$; samples $M$; task prompts $\{p^t\}$; weight $\lambda$

1: **Init:** For all $m \in \{1, \ldots, M\}$, set $(l^0_{(m)}, e^0_{(m)}) \leftarrow (\varnothing, \varnothing)$; $\mathcal{C}_{\min} \leftarrow \infty$
2: **for** $t = 1$ to $T_{\text{refine}}$ **do**
3:     **for** $m = 1$ to $M$ **do**
4:         $(l^t_{(m)}, e^t_{(m)}) \sim \pi_\theta\left(\cdot \,\middle|\, [x, l^{t-1}_{(m)}, e^{t-1}_{(m)}, p^t]\right)$
5:         $\mathcal{H}^{(m)}_{\text{det}} \leftarrow \text{Entropy}\big(\pi_\theta(l \,|\, [x, l^{t-1}_{(m)}, e^{t-1}_{(m)}, p^t])\big)$
6:         $\mathcal{H}^{(m)}_{\text{exp}} \leftarrow \text{MeanTokenEntropy}\big(\pi_\theta(e \,|\, [x, l^{t-1}_{(m)}, e^{t-1}_{(m)}, p^t])\big)$
7:         $\mathcal{C}^{(m)} \leftarrow \mathcal{H}^{(m)}_{\text{det}} + \lambda \, \mathcal{H}^{(m)}_{\text{exp}}$
8:     **end for**
9:     $m^\star \leftarrow \arg\min_m \mathcal{C}^{(m)}$
10:     If $\mathcal{C}^{(m^\star)} \geq \mathcal{C}_{\min} - \epsilon$, **break**
11:     Else $\mathcal{C}_{\min} \leftarrow \mathcal{C}^{(m^\star)}$
12: **end for**
13: **return** $(\hat{l}, \hat{e}) \leftarrow (l^t_{(m^\star)}, e^t_{(m^\star)})$

---

available supervision:

$$\max_\theta \quad \underbrace{\mathbb{E}_{(x,l) \sim D_{\text{sparse}}} \mathbb{E}_{(l^t, e^t) \sim \pi_\theta(\cdot \,|\, [x, l^{t-1}, e^{t-1}, p^t])} \big[\mathbb{1}(l^t = l)\big]}_{\text{sparse data: supervise detection only}} \tag{1}$$

$$+ \underbrace{\mathbb{E}_{(x,l,e) \sim D_{\text{dense}}} \mathbb{E}_{(l^t, e^t) \sim \pi_\theta(\cdot \,|\, [x, l^{t-1}, e^{t-1}, p^t])} \big[\mathbb{1}(l^t = l) + \mathbb{1}(e^t = e)\big]}_{\text{dense data: supervise detection \& reasoning}}. \tag{2}$$

During training, we construct sequential decision paths using both the sparse dataset $D_{\text{sparse}}$ and the dense dataset $D_{\text{dense}}$ to fine-tune the VLM. Our training procedure consists of two phases: offline imitation and online refinement, as detailed in Algorithm 1.

**Offline Imitation.** We begin by warm-starting the VLM using both sparse and dense datasets. In the first stage, the model is fine-tuned to predict detection and reasoning outputs without conditioning on any prior rounds, i.e., $(l^1, e^1) \sim \pi_\theta(\cdot \,|\, [x, \varnothing, \varnothing, \varnothing])$. This provides a baseline initialization where supervision is applied directly to the first-step predictions. In the second stage, we leverage the dense dataset to provide conditional expert transitions. Specifically, we supervise predictions conditioned on the expert outputs, $(l^1, e^1) \sim \pi_\theta(\cdot \,|\, [x, l, e, p^t])$, which encourages the model to preserve correct prior predictions and maintain cross-task consistency. In practice, for each sample we randomly select one task to train and mask out its ground truth input while retaining other inputs. This prevents the model from copying the task input and allows it to focus on cross-task and visual information. Together, these two stages ensure that the model learns to produce valid single-round predictions and to align its outputs with expert demonstrations when conditioning on prior task results.

**Online Refinement.** To address the mismatch between expert demonstrations and the model's own behavior, we further fine-tune the model with online rollouts from the policy. For each training example, we roll out the policy for $T$ rounds. Losses are computed against available supervision: detection is always supervised (on both $D_{\text{sparse}}$ and $D_{\text{dense}}$), while reasoning is supervised only on $D_{\text{dense}}$. On $D_{\text{sparse}}$, reasoning is generated but not penalized. This procedure allows the model to refine over its own trajectories while leveraging the ground truth whenever available. For both offline and online imitation, detection is trained with binary cross-entropy (BCE) loss, and the reasoning is trained with next-token prediction (NTP) loss.

### 4.3 ITERATIVE SELF-REFINEMENT FOR INFERENCE

At inference time, ARMOR generates multiple refinement trajectories and uses entropy as a self-certainty metric (Kang et al., 2025), enabling accurate detection together with open-ended, human-like reasoning. Starting from $(l^0, e^0) = (\varnothing, \varnothing)$, for each round $t$ and trajectory $m$, the model predicts

$$(l^t_{(m)}, e^t_{(m)}) \sim \pi_\theta\left(\cdot \,\middle|\, [x, l^{t-1}_{(m)}, e^{t-1}_{(m)}, p]\right).$$

We compute detection entropy $\mathcal{H}_{\text{det}}$ and reasoning entropy $\mathcal{H}_{\text{reason}}$ conditioned on prior outputs, and form a combined entropy score:

$$\mathcal{C}^{(m)} = \mathcal{H}_{\text{det}}^{(m)} + \lambda \, \mathcal{H}_{\text{reason}}^{(m)}.$$

We run refinement until the best trajectory no longer reduces uncertainty, i.e., when its entropy score stops decreasing beyond a tolerance $\epsilon$. Entropy serves a crucial role in this process: it determines when refinement terminates and also guides sample selection by identifying the most confident trajectory. The final output $(\hat{l}, \hat{e})$ is taken from the trajectory with the lowest entropy score (highest confidence) at termination. A detailed algorithm for inference is provided in Algorithm 2.

## 5 EXPERIMENT RESULTS

We evaluate our method against baselines across four diverse failure datasets spanning household and industrial warehouse environments. Our goal is to answer the following research questions:

1. Does ARMOR improve failure detection accuracy over prior works?
2. Does ARMOR offer better failure reasoning over prior works?
3. Does ARMOR work when sparse labels come from a different environment?
4. How does each component of ARMOR affect performance?
5. How does refinement improve the model's prediction over time?

Table 1: **Quantitative Results on Failure Detection and Reasoning.** Metrics include detection accuracy and reasoning quality (LLM Fuzzy and ROUGE$_L$). Our method achieves higher performance across different domains, accurately detecting failures and producing high quality reasoning.

| Models | Evaluation Datasets | Detect Acc. ↑ | LLM Fuzzy ↑ | ROUGE$_L$ ↑ |
|---|---|---|---|---|
| Qwen2.5-VL | RLBench | 0.376 | 0.255 | 0.353 |
| | Sparrow | 0.453 | 0.240 | 0.137 |
| | Maniskill | 0.548 | 0.268 | 0.112 |
| | ARMBench | 0.500 | 0.349 | 0.208 |
| Cosmos-Reasoning | RLBench | 0.317 | 0.220 | 0.140 |
| | Sparrow | 0.510 | 0.346 | 0.311 |
| | Maniskill | 0.442 | 0.359 | 0.207 |
| | ARMBench | 0.480 | 0.503 | 0.480 |
| LLaVA-NeXT | RLBench | 0.067 | 0.346 | 0.032 |
| | Sparrow | 0.500 | 0.008 | 0.078 |
| | Maniskill | 0.010 | 0.286 | 0.042 |
| | ARMBench | 0.500 | 0.000 | 0.067 |
| Claude-3.7 | RLBench | 0.420 | 0.372 | 0.336 |
| | Sparrow | 0.517 | 0.213 | 0.138 |
| | Maniskill | 0.538 | 0.360 | 0.133 |
| | ARMBench | 0.590 | 0.494 | 0.387 |
| Claude-3.7 (3-shot) | RLBench | 0.561 | 0.473 | 0.526 |
| | Sparrow | 0.650 | 0.407 | 0.458 |
| | Maniskill | 0.625 | 0.398 | 0.212 |
| | ARMBench | 0.650 | 0.685 | **0.725** |
| SFT-D | RLBench | 0.640 | 0.460 | 0.606 |
| | Sparrow | 0.523 | 0.278 | 0.274 |
| | Maniskill | 0.788 | 0.644 | 0.743 |
| | ARMBench | 0.640 | 0.609 | 0.618 |
| SFT-S+D | RLBench | 0.726 | 0.550 | 0.646 |
| | Sparrow | 0.620 | 0.245 | 0.257 |
| | Maniskill (R→M) | 0.490 | 0.177 | 0.381 |
| | ARMBench (S→A) | 0.495 | 0.007 | 0.249 |
| ARMOR (Ours) | RLBench | **0.917** | **0.718** | **0.802** |
| | Sparrow | **0.733** | **0.503** | **0.514** |
| | Maniskill (R→M) | **0.990** | **0.673** | **0.851** |
| | ARMBench (S→A) | **0.725** | **0.698** | 0.721 |

Table 2: Ablation of ARMOR's components. Each row shows the impact of different stages on detection accuracy and reasoning quality. ARMOR combines all components to achieve the best overall performance.

| Ablation | Offline Warmup | Offline Expert Condition | Online Imitation | Refinement (Inference) | Detection / Reasoning |
|---|---|---|---|---|---|
| Multitask Prediction | ✓ | ✗ | ✗ | ✗ | 0.897 / 0.460 |
| Refinement Only | ✓ | ✗ | ✗ | ✓ | 0.803 / 0.488 |
| Offline Imitation Only | ✓ | ✓ | ✗ | ✓ | 0.853 / 0.658 |
| Online Imitation Only | ✗ | ✗ | ✓ | ✓ | 0.850 / 0.683 |
| **ARMOR (ours)** | ✓ | ✓ | ✓ | ✓ | **0.917 / 0.718** |

**Datasets.** We evaluate on four datasets across different domains. The first two come from the AHA paper (Duan et al., 2025): RLBench-Fail and Maniskill-Fail. These contain simulated tabletop manipulation tasks such as "putting the trash in the bin" or "inserting the peg in the hole," where failures are induced by perturbing a scripted policy with specific failure modes (e.g., "the robot did not close its gripper"). Following AHA's official codebase, we generate the failure datasets and split them into dense labels (with failure modes) and sparse labels (without). We refer readers to the AHA paper for further details. Sparrow-Fail and ARMBench (Mitash et al., 2023) are from real warehouse operations; both are collected with a robotic manipulator transporting objects between containers with heterogeneous contents. Failures in these datasets are more diverse and nuanced, with sparse labels obtained from system logs and dense labels from human annotators.

**Baselines.** We compare against three state-of-the-art open-source VLMs—Qwen2.5-VL (Bai et al., 2025), Cosmos-Reasoning (Azzolini et al., 2025), and LLaVA-NeXT (Zhang et al., 2024a)—as well as one proprietary model, Claude-3.7 (with few-shot prompting). For fine-tuning baselines, following prior work (Duan et al., 2025), we evaluate supervised fine-tuning on dense data only (SFT-D), on both dense and sparse data (SFT-S+D). Unless otherwise specified, all open-source VLMs and ARMOR use 7B model variants.

**Metrics.** We report Binary Success Rate for detection, LLM Fuzzy Match (semantic similarity) and ROUGE-L (longest common sequence) for reasoning. Differ from AHA (Duan et al., 2025), we compute the LLM fuzzy score only on the reasoning part to prevent detection result from affecting reasoning metrics. Details on the datasets, baselines, and metrics are provided in Appendix B

## 5.1 FAILURE DETECTION AND REASONING PERFORMANCE

[Q1 & Q2] Table 1 shows that ARMOR achieves the highest scores in both binary detection and reasoning quality across all datasets. Prior fine-tuning approaches suffer from the dataset imbalance: SFT-D only leverages the limited dense data and produces suboptimal results, and SFT-S+D treats dense and sparse data as the standard next token prediction task, which often causes the model to overfit to the sparse data while failing to produce meaningful reasoning, leading to inferior performance. Open-source VLMs such as Qwen2.5-VL and Cosmos-Reasoning cannot perform beyond random guessing (50% accuracy), suggesting that generic instruction-tuned VLMs requires alignment for failure reasoning. Proprietary VLMs like Claude achieve stronger overall performance, especially with few-shot prompting, but still fall short in performance to ARMOR in most tasks.

[Q3] We combine sparse data from RLBench and dense data from Maniskill and evaluate in Maniskill (R→M) to study small distribution shifts, where they differ in embodiment and workspace setup. For large distribution shifts, we combine sparse data from Sparrow and dense data from ARMBench (S→A), where they differ in both workspace setup and failure reasoning. In R→M, ARMOR improves over SFT-D by **+25.6%** in detection and maintains high reasoning quality. SFT-S+D reasoning collapses from 0.644 to 0.177, showing that naively mixing dense and sparse data prevents generalization. In the harder S→A setting, ARMOR outperforms SFT-D by **+13.3%** in detection and **+14.6%** in reasoning, and substantially exceeds SFT-S+D. These results show that our model consistently improves both detection and reasoning quality across multiple reasoning metrics.

## 5.2 ABLATION STUDIES

[Q4] We consider the following ablations to our method evaluated in RLBench. **Multitask Prediction** employs two heads for prediction and only fine-tunes on unconditioned offline data. This isolates the benefit of our multi-task architectural design without conditional training or refinement. **Refinement Only** uses ARMOR's architecture to only perform self-refinement at inference, removing training (except for warm-up) from our algorithm. **Offline Imitation** removes training on online transitions, while **Online Imitation Only** only trains on online transitions. All ablation models are

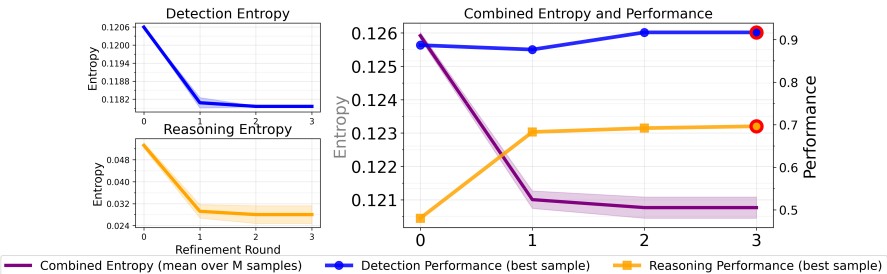

(a) Effect of refinement rounds on entropy and performance.

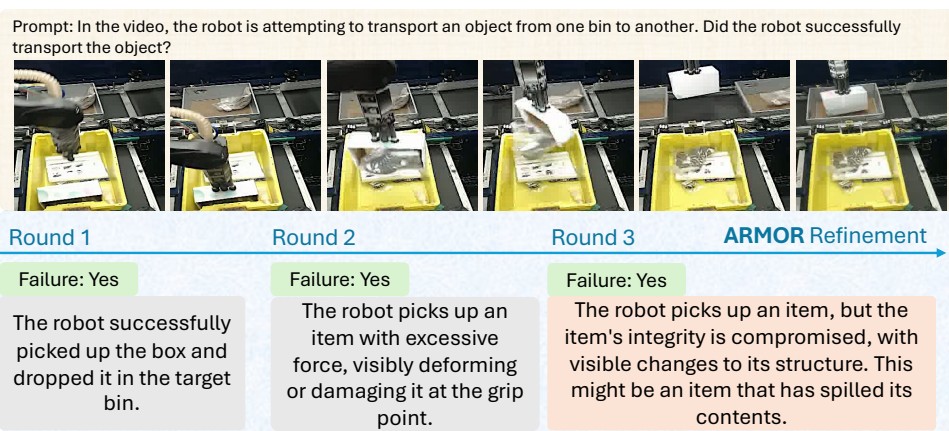

(b) Example of ARMOR's multi-round refinement process.

Figure 3: Refinement analysis in ARMOR. (a) Effect of refinement rounds on entropy and performance: detection and reasoning entropy decrease steadily across rounds, while combined plots show that refinement improves both detection accuracy and reasoning quality (values computed over 300 test datapoints). (b) Example of ARMOR's multi-round refinement process, where predictions are iteratively updated to improve the quality and consistency between detection and reasoning outputs.

trained for the same total number of epochs as ARMOR. When an ablation excludes certain training stages, we add additional epochs to the remaining stages to ensure fair comparison. We visualize each ablation's training stages and their performance in Table 2. Multitask Prediction already achieves strong detection accuracy (0.897) but suffers from poor reasoning quality (0.460), confirming that architectural design alone is insufficient. Refinement Only slightly improves reasoning (0.488) but reduces detection accuracy (0.803), suggesting that multi-task conditional training is necessary. Offline Imitation and Online Imitation both improve reasoning quality compared to the previous variants, with online data contributing more strongly to reasoning. Finally, combining all components in ARMOR yields the best performance, highlighting the benefits of multi-task design, refinement, and offline and online training.

**Number of Refinement Rounds.** We ablate the number of refinement rounds and track both performance and entropy using test data in RLBench. We run inference with 4 seeds and report the mean and standard deviation for each refinement round in Table 3. This demonstrates that the refinements consistently improve performance with decreasing variance.

In Figure 3a, entropy for detection and reasoning decreases across rounds and stabilizes during refinement (left), while the combined entropy $\mathcal{C}$ alongside task performance (right) shows that detection accuracy improves with additional refinement, and reasoning exhibits significant gains after the first round. Together, these results confirm that ARMOR's iterative refinement reduces uncertainty and enhances both detection and reasoning quality. We provide additional ablation study results in Appendix C.1.

Table 3: ARMOR's reasoning performance across refinement rounds. Mean and std are reported across 4 seeds. Each round shows improvements with decreasing variance, demonstrating consistent gains.

| Refinement Rounds | Reasoning (LLM Fuzzy) |
|---|---|
| Round 0 | 0.475±0.016 |
| Round 1 | 0.676±0.025 (+42.4%) |
| Round 2 | 0.703±0.013 (+4.0%) |
| Round 3 | **0.717±0.002** (+2.0%) |

## 5.3 REFINEMENT PROCESS

We first address [Q5] by showing that performance improves over refinement rounds in Figure 3a. Next, we provide qualitative results in Figure 3b, showing how ARMOR iteratively improves its predictions. In the first round, the model produces incorrect reasoning while correctly detecting the failure. In later rounds, it first adjusts the reasoning to stay consistent with the detection result (failure) and then refines it into an accurate account of the failure. This process demonstrates how multi-round refinement reduces inconsistencies and enhances reasoning quality. In general, we observe that reasoning results tend to be corrected based on detection, since detection is generally more reliable (as more data is available for detection), though there are also cases where both predictions are corrected through successful refinement. Additional examples are provided in Appendix C.2.

## 5.4 INFERENCE COST

We evaluate the computational overhead of ARMOR's refinement process on 8×A100 GPUs (40GBs of memory each) using a batched implementation where multiple rollouts increase only memory usage, not inference time. Table 4 shows memory and time usage across refinement rounds for a single question, where round 0 refers to the base model's (Bai et al., 2025) latency. ARMOR adds minimal computational overhead compared to the base model—approximately 1s increase in wall clock time per round, with modest memory overhead (from 3.48GB to 6.31GB per GPU). To balance performance and efficiency, one can choose to use fewer rounds of refinement (e.g., 1 round refinement already improves reasoning by 40%).

Table 4: Memory usage and wall clock time across refinement rounds. ARMOR adds minimal computational overhead, with approximately 1s increase per round and modest memory growth.

| Refinement Rounds | Avg. Memory Usage per GPU (GB) | Wall Clock Time (s) |
|---|---|---|
| Round 0 | 3.48 | 7.95 |
| Round 1 | 5.84 | 9.30 |
| Round 2 | 6.12 | 10.50 |
| Round 3 | 6.31 | 10.95 |

## 6 CONCLUSION

We introduced ARMOR, a new method for robotic failure detection and reasoning based on iterative multi-task refinement. ARMOR learns from both sparse labels and dense annotations, handles heterogeneous supervision, and extends VLMs to open-ended reasoning beyond predefined failure modes. Through experiments across simulated and real-world environments, we highlighted that ARMOR achieves state-of-the-art performance in both detection and reasoning. For future work, we plan to incorporate richer training signals such as task rewards and human preferences to further enhance reasoning quality and alignment with human. We also plan to extend ARMOR to other synergistic tasks, such as predicting safety violations and recovery strategies, which could enhance its applicability in real-world robotics. Finally, integrating modalities beyond vision, including force-torque sensing and proprioception, may capture a wider distribution of failures in the real world.

**Broader Impact.** Robotic failure detection and reasoning plays a critical role in enabling autonomous systems to operate safely and effectively in real-world environments. While binary failure detection alone may suffice for controlled settings with known failure modes, our work focuses on failure reasoning, which can provide interpretable feedback for robot correction and human intervention. This is essential is crucial to improve robot policy and reduce the need for expert supervision during deployment. Furthermore, manipulation tasks increasingly involve diverse objects, environments, and failure modes that benefit from the generalization capabilities of VLMs. As VLMs continue to advance, they hold promise for enabling more generalizable and interpretable failure reasoning in autonomous systems, and our work contributes to this direction by demonstrating how to effectively leverage heterogeneous supervision for better performance.

ETHICS STATEMENT

We have adhered to ethical research practices and legal standards. Our study advances methods for robotic failure detection and reasoning using video datasets. We use simulated or warehouse-collected data without personal information. The research aims to improve the safety, transparency, and reliability of autonomous systems, aligning with the principles of responsible stewardship and minimizing potential harm. We acknowledge that over-reliance on imperfect reasoning models in safety-critical settings may pose risks, and emphasize that our method should complement, not replace, rigorous monitoring and human oversight.

REPRODUCIBILITY STATEMENT

To ensure the reproducibility of our work, we include detailed descriptions of the implementation of our algorithm in Appendix A, datasets in Appendix B.1, and full prompts and evaluation protocol in Appendix B.4. We will release open-sourced datasets and code upon publication.

ACKNOWLEDGMENTS

The project was developed during CQ's internship at Amazon Robotics. AZ and CQ were funded by NSF 2340651, NSF 2402650, DARPA HR00112490431, and ARO W911NF-24-1-0193.

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

# Appendix

## THE USE OF LARGE LANGUAGE MODELS (LLMS)

We have used LLMs for polishing the writing of the paper. We have also employed an LLM to compute one of the evaluation metrics (LLM Fuzzy Matching, detailed in Appendix B.3). LLM does not serve as a major contributor to this paper, as all of the ideas are developed and implemented by human authors.

## A    IMPLEMENTATION DETAILS

### A.1    MODEL ARCHITECTURE

We use Qwen2.5-VL (Bai et al., 2025) as the VLM backbone for ARMOR. For detection, we attach a lightweight binary classification head. The head takes mean-pooled features from the first four LM decoder layers, projects them, cross-attends with a learnable `[CLS]` token, and decodes with an MLP to produce logits. Reasoning is generated using the base model's auto-regressive decoder. Both heads share the same vision encoder and LM decoder parameters. Conditioning is injected through prompts such as ``Given the previous detection is ...''  or ``Given the previous reasoning is ...'', ensuring that each round depends on the prior outputs. We provide a schematic of the architecture in Figure 4 and pseudo-code for the classifier head below:

```python
class VLClassifierHead(nn.Module):
    def __init__(self, hidden_size=4096, num_classes=2, dropout=0.1):
        super().__init__()
        self.cls_token = nn.Parameter(torch.zeros(1, 1, hidden_size))

        # Feature projection
        self.projector = nn.Sequential(
            nn.Linear(hidden_size, hidden_size),
            nn.GELU(),
            nn.Dropout(dropout),
            nn.Linear(hidden_size, hidden_size),
        )

        # Aggregation via attention
        self.agg_attn = nn.MultiheadAttention(
            embed_dim=hidden_size,
            num_heads=8,
            batch_first=True
        )

        # Output classifier MLP
        self.out_mlp = nn.Sequential(
            nn.Linear(hidden_size, hidden_size),
            nn.GELU(), nn.Dropout(dropout),
            nn.Linear(hidden_size, hidden_size // 2),
            nn.GELU(), nn.Dropout(dropout),
            nn.Linear(hidden_size // 2, num_classes)
        )

    def forward(self, features):
        B = features.size(0)
        cls = self.cls_token.expand(B, -1, -1)
        features = self.ln(self.projector(features))
        attn, _ = self.agg_attn(cls, features, features)
        x = self.out_ln(attn.squeeze(1))
        return self.out_mlp(x)
```

## A.2 TRAINING AND INFERENCE

We leverage an open-source codebase (Lee, 2024) to run our fine-tuning experiments. During offline conditional training, when the model is given ground-truth prior predictions $(l, e)$ as inputs, we randomly mask out one of the predictions so that it also learns to refine from imperfect contexts rather than overfitting to the fixed-point supervision. Specifically, we first randomly select a task, and then we mask out the ground truth input for that same task while retaining the ground truth from the other task. For example, when predicting reasoning, we mask the ground truth reasoning input but keep the detection input, allowing the model to leverage cross-task information along with visual context. For video-based datasets (Sparrow and ARMBench), we use a global batch size of 16, while for collaged image datasets (RLBench and Maniskill) we use a batch size of 64. All models are trained for 3 epochs for offline phase and 10 epochs online phase on 8×H100 GPUs. We use AdamW with weight decay 0.1 and cosine learning rate scheduling with a warmup ratio of 0.03. The base learning rate for the LM decoder and classification head is set to $1 \times 10^{-5}$, while the vision encoder is trained with a smaller learning rate of $2 \times 10^{-6}$. For refinement, we set $\lambda = 0.1$ to place greater weight on detection entropy, since the detection head benefits from supervision on both $D_{\text{sparse}}$ and $D_{\text{dense}}$ and is thus more reliable than the explanation predictions. We summarize the hyperparameters in Table 5.

Table 5: **Hyperparameters for training and inference.**

| Hyperparameter | Value |
|---|---|
| Global batch size (videos) | 16 |
| Global batch size (collaged images) | 64 |
| Offline epochs | 3 |
| Online epochs | 10 |
| Horizon $T$ | 3 |
| Frames per second (FPS) | 5.0 |
| Learning rate (LM + classifier) | $1 \times 10^{-5}$ |
| Learning rate (vision encoder) | $2 \times 10^{-6}$ |
| Weight decay | 0.1 |
| Warmup ratio | 0.03 |
| LR scheduler | Cosine |
| Dropout | 0.1 |
| Refinement weight $\lambda$ | 0.1 |
| Refinement Samples $M$ | 3 |
| Max Refinement rounds $T_{refine}$ | 4 |
| Entropy tolerance $\epsilon$ | $1 \times 10^{-4}$ |
| Temperature $\tau$ | 0.7 |
| Top p | None |
| Number of beams | 1 |

## B EXPERIMENT DETAILS

| Dataset | Domain | Sparse | Dense | Test | Unseen task | Unseen workspace | Unseen failure |
|---|---|---|---|---|---|---|---|
| RLBench | Simulation | 8000 | 1000 | 300 | ✓ | ✗ | ✗ |
| Maniskill (R → M) | Simulation | 0 | 600 | 100 | ✓ | ✓ | ✗ |
| Sparrow | Real world | 3000 | 600 | 300 | ✗ | ✗ | ✓ |
| ARMBench (S → A) | Real world | 0 | 600 | 200 | ✗ | ✓ | ✓ |

Table 6: **Details of datasets.** Dataset domains, number of sparse and dense samples used for training, and types of distribution shifts between train and test data.

### B.1 DATASETS

Table 6 provides an overview of dataset sizes and domains. Below we describe each dataset and its generation procedure in detail.

**RLBench** (James et al., 2020) is a simulator on top of which AHA (Duan et al., 2025) developed a wrapper to generate a set of predefined failure modes. The implementation contains diverse tasks

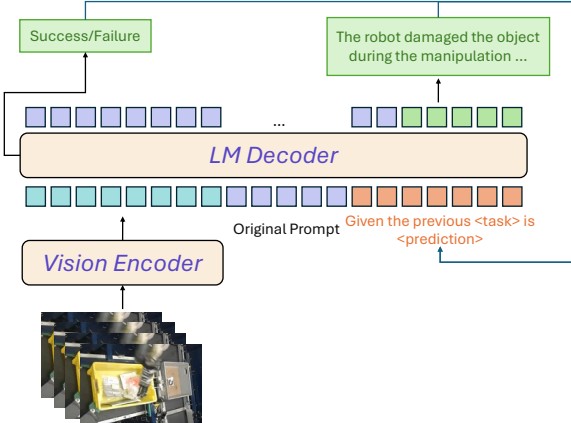

Figure 4: **ARMOR model architecture.** We select the intermediate layer representation from the LM decoder of Qwen2.5-VL (Bai et al., 2025) and attach a classification head for detection, while using the original LM decoder for reasoning. Conditioning prompts describe the previous outputs for each task, enabling iterative multi-task refinement.

such as "putting the trash in the bin" or "inserting the peg in the hole," where failures are induced by perturbing a scripted policy with specific failure modes (e.g., "the robot did not close its gripper"). Following AHA, we construct input images by combining multiple frames across different time steps and camera views into a matrix: the top row shows the front view, the middle the wrist view, and the bottom the overhead view, with timesteps annotated left to right. Using AHA's code, we generated our own RLBench dataset across 10 tasks with 8000 sparse samples and 1000 dense samples. We evaluate on three unseen tasks.

**Maniskill** (Mu et al., 2021) is a simulator for table-top manipulation, also used in AHA. This dataset differs from RLBench in its tasks and robot embodiment (Sawyer robot). We follow the same multi-view image construction protocol as RLBench and generate 600 dense samples for 5 tasks, which are combined with the sparse samples from RLBench to create a training dataset for the transfer setting (R → M).

**ARMBench** (Mitash et al., 2023) is collected in a real warehouse with a robotic manipulator transporting objects between containers with heterogeneous contents. Failures include object defects such as books opening or items spilling their contents. We directly use the raw video feed resized to $320 \times 320$ and sampled at 5 FPS. Sparse samples are obtained from system logs, and dense samples are provided by human annotators. We use 600 dense samples for training and combine them with sparse samples from Sparrow to create a dataset for the transfer setting (S → A).

**Sparrow** is a custom dataset similar to ARMBench but differs in workspace setup, bin locations, and camera angles. It also contains more diverse failures and annotations. As with ARMBench, we use the resized $320 \times 320$ video feed sampled at 5 FPS. Failures are annotated in more nuanced ways, with sparse samples obtained from system logs and dense samples from human annotators. We use 3000 sparse samples and 600 dense samples from Sparrow.

## B.2 BASELINES

**Open-source VLMs.** We compare against three state-of-the-art open-source vision-language models (VLMs): Qwen2.5-VL (Bai et al., 2025), Cosmos-Reasoning (Azzolini et al., 2025), and LLaVA-NeXT (Zhang et al., 2024a). Qwen2.5-VL is a recent large-scale VLM with strong visual grounding and instruction-following capability. Cosmos-Reasoning extends standard VLMs with improved chain-of-thought prompting for reasoning-heavy tasks. LLaVA-NeXT is an efficient vision-language model optimized for video and temporal grounding.

**Proprietary VLM.** We also evaluate Claude-3.7, a proprietary model accessed through its API. We apply few-shot prompting using structured templates that encourage the model to output both reasoning and a binary decision. The full prompt examples are provided in Appendix B.4.

**Fine-tuning methods.** Following prior work (Duan et al., 2025), we include supervised fine-tuning baselines: **SFT-D:** fine-tuning on dense data only. **SFT-S+D:** fine-tuning on both sparse and dense data jointly, where we simply has empty strings between `<think></think>` for sparse data. All models are trained for the same number of epochs (13) as our method, using the Qwen2.5-VL (Bai et al., 2025) backbone and identical hyperparameters.

## B.3 EVALUATION METRICS

We employ three complementary metrics to assess both failure detection and reasoning quality.
**Binary Success Rate.** For detection accuracy, we compute the binary success rate by directly comparing predicted labels $\hat{l}$ with ground-truth labels $l$:

$$\text{Accuracy} = \frac{1}{N} \sum_{i=1}^{N} \mathbb{1}\{\hat{l}_i = l_i\},$$

where $N$ is the number of test samples. This provides a direct measure of failure detection.

**LLM Fuzzy Matching.** We employ LLM Fuzzy Matching, which leverages an external language model (Claude-3.7-Sonnet) to judge semantic similarity between generated explanations and references. The external model evaluates whether two explanations are semantically equivalent, even when their surface forms differ.

**ROUGE-L.** (Ganesan, 2006) To complement LLM Fuzzy Matching, we compute the ROUGE-L score to evaluate textual reasoning quality. ROUGE-L measures the similarity between a generated reasoning $y$ and a human-annotated reference $y^*$ by focusing on the longest common subsequence (LCS). Let $m$ and $n$ be the lengths of $y$ and $y^*$. The score $\text{ROUGE}_L$ is defined as:

$$F_\beta = \frac{(1 + \beta^2) \cdot P \cdot R}{R + \beta^2 \cdot P},$$

where $\beta$ is typically set to 1. This metric captures fluency and informativeness by rewarding longer, in-sequence matches rather than isolated $n$-grams.

## B.4 PROMPTS AND EVALUATION PROTOCOL

**General Prompts.** For all baseline models, we apply the following prompt. The prompt explicitly asks the VLM to provide both reasoning and a binary decision in the following format:

```
{"question_id": "0",
 "system_prompt": "You are observing a video showing embodied agents
 ↪  (robots or humans) demonstration.
Please answer the question in the following format:
<think>your reasoning</think>
<answer>Yes/No</answer>.",
"user_prompt": "In the video, the robot is attempting to transport an
↪  object from one bin to another.\nDid the robot successfully
↪  transport the object?"
```

**ARMOR Prompts.** Since ARMOR uses multi-task prediction, each task is associated with a slightly different prompt, optionally including conditioning prompts from previous rounds.

```
######## DETECTION ########
In the video, the robot is attempting to transport an object from one
↪  bin to another.
Did the robot successfully transport the object?
[Conditioning: "Given the previous detection is ...",
 "Given the previous reasoning is ..."]
Please give a Yes/No answer.

######## REASONING ########
In the video, the robot is attempting to transport an object from one
↪  bin to another.
```

```
Did the robot successfully transport the object?
[Conditioning: "Given the previous detection is ...",
 "Given the previous reasoning is ..."]
Please give your reasoning.
```

**Few Shot Prompts.** For the proprietary baseline Claude-3.7, we query the model through its API using few-shot prompting. The general prompt is the same for Claude and others. We additionally provide the full few-shot prompt used in evaluation below:

```
{"question_id": "0",
 "system_prompt": "You are observing a video showing embodied agents
 ↪  (robots or humans) demonstration.
Please answer the question in the following format:
<think>your reasoning</think>
<answer>Yes/No</answer>.",
"user_prompt": "For the given sub-tasks, first determine it has succeed
↪  by choosing from [\"yes\", \"no\"], and then explain the reason why
↪  the current sub-tasks has failed.
Here are some examples:

Example 1:
Question: In the image ... robot arm performing pick up the red cube.
<think>The robot succeeded at the sub-task.</think>
<answer>yes</answer>

Example 2:
Question: In the image ... robot arm performing plug the white charger
↪  in the socket.
<think>The robot succeeded at the sub-task.</think>
<answer>yes</answer>

Example 3:
Question: In the image ... robot arm performing pick up the red cube.
<think>No, the robot gripper moved to the desired position with an
↪  offset along the y direction.</think>
<answer>no</answer>

Now answer the following question:
Question: In the image ... robot arm performing pick up the red cube."}
```

**Evaluation protocol.** For ARMOR, we use the classifier head's output for detection prediction and the explanation head's output for reasoning, avoiding the need to do regular expression matching for answer extraction. For baselines, following AHA (Duan et al., 2025), we extract the binary decision from the `<answer>` field to compute detection accuracy, and evaluate the free-form reasoning from the `<think>` field separately using LLM Fuzzy Match and ROUGE$_L$. If the model does not follow the required output format (i.e., if the two fields cannot be extracted via regular expression matching), we treat the entire output as the explanation and set detection correctness to false. This evaluation procedure is somewhat brittle, since it cannot distinguish between a model that fails to follow formatting instructions and one that consistently outputs the wrong answer. To mitigate this issue, all of our test datasets are balanced to contain an equal number of success and failure cases.

## C  ADDITIONAL RESULTS

**Example of inconsistent prediction from Qwen2.5-VL (Bai et al., 2025) model.** The following example highlights the inconsistency: the baseline model's reasoning states that the robot successfully transported the object, yet its binary decision is No (failure). Such contradictions are common when failure detection and reasoning are treated as a single free-form generation task, motivating ARMOR's explicit multi-task refinement design.

```
<think>The robot's arm is positioned above the blue bin, and it appears
↪   to be in the process of picking up an object. The object is not
↪   visible in the image, but based on the robot's posture and the
↪   context of the task, it seems that the robot has
↪   successfully transported the object from the left bin to the right
↪   bin.</think>
<answer> No </answer>

######### Ground truth #########

<think>The robot attempts to pick up an object from the source bin, but
↪   fails to successfully grasp it. I can see the suction gripper moving
↪   toward the target object and making contact with it. When the
↪   gripper attempts to lift, the object remains in the bin instead of
↪   being lifted. I notice the robot arm moving away without the
↪   intended item. Since the robot's task requires transferring objects,
↪   this failure prevents successful task completion."</think>
<answer>No</answer>
```

## C.1 ADDITIONAL ABLATION STUDIES

**Effect of Sparse-Dense Data Ratio.** We provide experiments showing performance across varying ratios of sparse to dense annotations on the Sparrow→ARMBench setting. We fix the amount of sparse data from Sparrow and vary the amount of dense data in ARMBench, as shown in Table 7. The results show that ARMOR is robust to up to 10× data imbalance between sparse and dense annotations (300 dense, 3000 sparse samples). However, further decreasing the dense samples causes reasoning performance to deteriorate. ARMOR scales well with increasing dense data, achieving significant performance gains when we decrease the ratio to 2 (1500 dense, 3000 sparse samples).

Table 7: Performance across varying sparse-dense data ratios. The column in parentheses represents the setting used in the main paper.

| Sparse / Dense Ratio | 2 | (5) | 10 | 30 |
|---|---|---|---|---|
| Detection / Reasoning | 0.813 / 0.831 | (0.725 / 0.698) | 0.640 / 0.609 | 0.620 / 0.427 |

**Effect of Model Size.** We evaluated ARMOR using the 32B version of Qwen2.5-VL (Bai et al., 2025) on Sparrow and observed consistent performance gains compared to the 7B version, as shown in Table 8. The results demonstrate that ARMOR's approach scales effectively with larger model capacity.

Table 8: Performance comparison across model sizes on Sparrow dataset.

| Model | Detection / Reasoning |
|---|---|
| Qwen2.5-VL 7B | 0.453 / 0.240 |
| Qwen2.5-VL 32B | 0.563 / 0.268 |
| ARMOR 7B (ours) | 0.733 / 0.503 |
| ARMOR 32B (ours) | **0.765 / 0.562** |

## C.2 MORE REFINEMENT EXAMPLES

**Example of refinement that corrects both detection and reasoning.** The following example shows how iterative refinement can recover from an initial misclassification (No) and inconsistent reasoning, converging to a consistent and correct prediction by the final round. The task execution frames are shown in Figure 5.

```
<prompt>In the video, the robot is attempting to pick up a hockey stick.
Frames are shown from multiple views (front, wrist, overhead) across
↪   timesteps. Did the robot succeed at the subtask?</prompt>
```

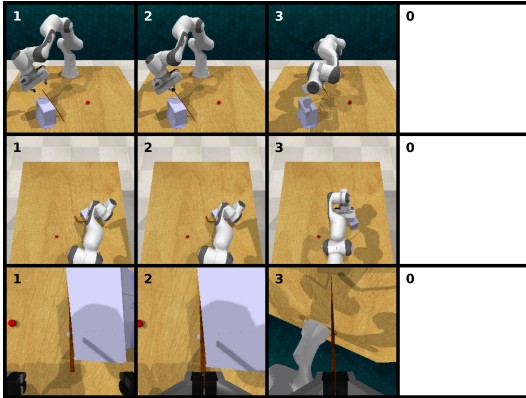

Figure 5: **Hockey-stick task from RLBench** Multi-view frames (front, wrist, overhead) across timesteps showing the robot attempting to pick up a hockey stick.

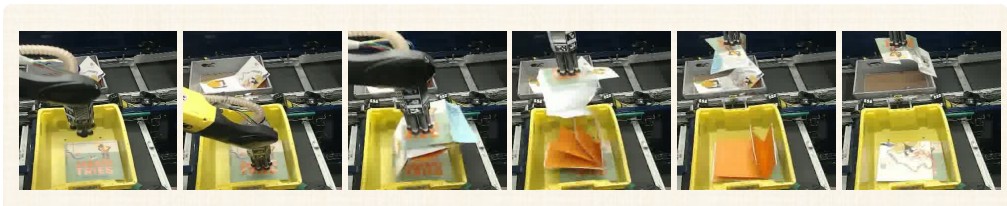

Figure 6: **ARMBench transport task.** The ground truth failure is item damage caused by detached book cover.

```
######### Ground truth #########
<think>The robot succeeded at the sub-task.</think>
<answer>Yes</answer>

######### Refinement History #########
Step 1:
  Detection = No
  Reasoning = The robot gripper moved with an offset along the y
  ↪  direction.

Step 2:
  Detection = Yes
  Reasoning = The robot fails to close the gripper.

Step 3 (Final Output):
  Detection = Yes
  Reasoning = The robot succeeded at the sub-task.
```

**Example of refinement failure: refinement does not converge to the correct reasoning.**

We use an example from ARMBench (depicted in Figure 6) to show a failure of ARMOR's refinement: while the binary prediction is correct (No), the reasoning drifts toward alternative but incorrect explanations (collision → misplacement) and never captures the true failure (item damage). Future work could incorporate structured failure attributes (e.g., "collision," "misplacement," "damage") as intermediate supervision in the form of reward signal to better constrain the model's reasoning.

```
<prompt>
In the video, the robot is attempting to transport an object from one
↪  bin to another. Did the robot successfully transport the object?
</prompt>
```

```
########## Ground truth ##########
<think>I can see visible damage to the item during the robot's handling
↪  process. There are clear signs of physical harm such as crushing,
↪  tearing, or deformation of the package. The item's integrity is
↪  compromised, with visible changes to its structure. The robot picked
↪  up the book, but the cover of the book fell off,  causing the book
↪  to be damaged. This damage reflects a critical handling failure that
↪  impacts product quality and customer satisfaction, as the item is no
↪  longer in its intended condition.</think>
<answer>No</answer>

########## Refinement History ##########
Round 1:
  Detection = No
  Reasoning = I observe the robot's arm or gripper making an unintended
  ↪  forceful contact with objects in the environment. The impact is
  ↪  clearly visible as the robot's smooth trajectory is disrupted when
  ↪  it strikes either the tote wall, other items, or surrounding
  ↪  structures.

Round 2:
  Detection = No
  Reasoning = I see that the robot has successfully picked up the item
  ↪  from the source location, but then places it in what appears to be
  ↪  the wrong destination. The robot arm moves to a location that
  ↪  doesn't match the expected placement target for this task.

Round 3 (Final Output):
  Detection = No
  Reasoning = I see that the robot has successfully picked up the item
  ↪  from the source location, but then places it in what appears to be
  ↪  the wrong destination. The robot arm moves to a location that
  ↪  doesn't match the expected placement target for this task.
```

