# OpenReview forum: "Self-Refining Vision Language Model for Robotic Failure Detection and Reasoning"
_ICLR.cc/2026/Conference — ICLR 2026 Poster_

### Official Review · Reviewer_2SV2 · 2025-10-24

**Soundness:** 3
**Presentation:** 2
**Contribution:** 2
**Rating:** 6
**Confidence:** 4

**Summary:**

The paper introduce ARMOR, an approach that jointly detects whether a robot execution succeed or failed and then provide explains and building on top of prior work like AHA, and they achieve this through iteratively refines both outputs over several rounds. It is designed for the realistic case where you have many binary success/failure labels but few natural‑language explanations. Training mixes offline imitation (warm‑up + conditioning on ground‑truth previous outputs) with online refinement rollouts; inference runs several refinement trajectories and selects the output with lowest entropy.

**Strengths:**

Strength of the paper:
-> The method demonstrated to be effiective and efficient, as on RLBench‑Fail & ManiSkill‑Fail (sim), Sparrow‑Fail & ARMBench (real). Detection is measured by Binary Success Rate; reasoning by LLM Fuzzy Match (judge: Claude‑3.7‑Sonnet) and ROUGE‑L; the fuzzy score is computed only on the reasoning text to avoid format bias.
-> The work also demonstrated that the explicitly target s the common regime with a lot of binary label data, and only a few explains which still able to achieve good performance.
->Gains hold from simulation to real warehouses and under transfer (R→M, S→A), where naïvely mixing sparse+dense (SFT‑S+D) often collapses reasoning.

**Weaknesses:**

-> Unreported runtime or inference time latency, as the inference does multiple rounds and samples, so it is good to know the wall-clock or throughput for this, cause it matters for robot deployment.

-> There are many runs on benchmarks from AHA, however there was not any demonstrate of how such VLM model could be used to help with downstream robotics manipulation tasks.

-> How well does this transfer to cross-embodiment to different single arm robot.

**Questions:**

Refer to the weakness for the questions.

---

> ### Author Response · Authors · 2025-11-14
> **Response to reviewer's comments**
>
> We appreciate the reviewer for their constructive feedback, and we are happy to address each concern below.
>
> ---
> **”Unreported runtime or inference time latency, as the inference does multiple rounds and samples, so it is good to know the wall-clock or throughput for this, cause it matters for robot deployment.”**
> * We tested inference on 8×A100 GPUs (40GB memory each) using a batched implementation where multiple rollouts increase only memory usage, not inference time. Memory and time usage across refinement rounds for a single sample are shown below, where round 0 corresponds to the base Qwen model latency. (table will be included in the revised paper):
> | Refinement Rounds | Avg. Memory Usage per GPU (GB) | Wall Clock Time (s) |
> |-------------------|-------------------|-------------------|
> | 0 | 3.48 | 7.95 |
> | 1 | 5.84 | 9.30 |
> | 2 | 6.12 | 10.50 |
> | 3 | 6.31 | 10.95 |
> * ARMOR adds minimal computational overhead compared to the base model—approximately 1s increase in wall clock time per round, with modest memory overhead (from 3.48GB to 6.31GB per GPU). To balance performance and efficiency, one can choose to use fewer rounds of refinement (e.g. 1 round refinement already improves reasoning by 40%)
> * ARMOR performs post-facto failure reasoning, which does not require real-time feedback. Since ARMOR is trained on execution videos, robots can complete a sequence of actions and then receive feedback from ARMOR. As tools like vLLM are still being actively developed, we believe VLM deployment optimization is an exciting area for future work to enable real-time applications.
>
> **"There are many runs on benchmarks from AHA, however there was not any demonstration of how such VLM models could be used to help with downstream robotics manipulation tasks."**
> * Our main focus is addressing failure detection and reasoning under heterogeneous data availability, making downstream task integration beyond the scope of this paper. However, since failure examples in RLBench and Maniskill are similar to those in AHA, ARMOR can serve as a drop-in replacement for AHA in their downstream applications. We believe that leveraging ARMOR's detailed failure reasoning output to provide actionable feedback for robot correction or human intervention is an exciting area for future work and will add this discussion to the paper.
>
> **"How well does this transfer to cross-embodiment to different single arm robots?"**
> * We test cross-embodiment transfer in our RLBench→Maniskill experiments. The robot embodiment differs between RLBench (Franka Panda) and Maniskill (Sawyer). We show a significant performance difference between our method and the baseline in cross-embodiment transfer (0.673 vs. 0.177 in reasoning quality). Results can be found in Table 1 of the paper under the R->M task.
> ---
>
> We believe these clarifications address the reviewer's concerns. We remain open to further discussion and hope the reviewer will consider these revisions favorably.

---

### Official Review · Reviewer_zLbK · 2025-10-29

**Soundness:** 2
**Presentation:** 3
**Contribution:** 3
**Rating:** 6
**Confidence:** 3

**Summary:**

The paper addresses failure detection and reasoning in robotic systems, which is indeed an important capability for enhancing the reliability of robotic systems. Prior work either treats this as closed-set classification or relies heavily on fully annotated reasoning labels. The authors propose ARMOR, a VLM that performs multi-round self-refinement to jointly predict binary failure detection and open-ended natural-language reasoning.

ARMOR is trained under heterogeneous supervision, combining large-scale sparse binary labels and a small number of dense reasoning annotations. The framework employs two task-specific heads (classification and text generation) trained via a combination of offline imitation and online refinement, allowing it to iteratively improve predictions conditioned on previous outputs. At inference time, multiple refinement trajectories are generated and scored using an entropy-based self-certainty metric, with the most confident prediction selected as output.

Experiments across four domains (RLBench, Maniskill, Sparrow, and ARMBench) show state-of-the-art performance, with up to 30% improvement in detection accuracy and 100% in reasoning quality (LLM fuzzy match) compared to prior baselines such as AHA and SFT variants. Ablations demonstrate that both multi-task design and iterative refinement contribute to performance gains.

**Strengths:**

- The paper addresses an under-explored but crucial problem, robotic failure detection and reasoning.

- The proposed multi-round adaptive refinement allows the model to iteratively improve its predictions, similar to human introspection, and provides more coherent reasoning explanations.

- The authors performed extensive experiments on four diverse robotic datasets (RLBench, ManiSkill, Sparrow, ARMBench), showing clear and consistent gains over strong baselines in both detection accuracy and reasoning quality.

- The authors conduct detailed ablation studies on the refinement rounds, task weighting, and uncertainty metrics, providing convincing evidence of each design component’s impact.

**Weaknesses:**

- How does the multi-round refinement introduce additional inference cost proportional to the number of rounds? It can be helpful to also include some latency or runtime statistics. In real-time robotic applications, providing timely feedback can be challenging without interrupting task execution. Similarly, the dependency on large pre-trained language models raises deployment challenges for real-time execution.

- For the failure detection of these safety-critical tasks, using these large models with even video inputs can be overkill compared to specialized settings. Since these manipulation tasks are not performed/validated in the wild, why can one not use more overfitting metrics instead of an open-set VLM if one cares more about the binary value of success or not than the detailed reasoning? Including some discussions would further improve the soundness of the paper.

- While the proposed multi-round self-refinement mechanism is well-motivated, the architectural backbone and training strategy primarily build upon the existing vision–language and instruction-tuning framework. The main novelty lies in integrating iterative refinement and multi-task learning, which, though valuable, represents an incremental advance rather than a new paradigm.

**Questions:**

See the questions above.

---

> ### Author Response · Authors · 2025-11-14
> **Response to reviewer's comments (1/2)**
>
> We appreciate the reviewer for their constructive feedback, and we are happy to address each concern below.
>
> ---
> **"How does the multi-round refinement introduce additional inference cost proportional to the number of rounds? It can be helpful to also include some latency or runtime statistics. In real-time robotic applications, providing timely feedback can be challenging without interrupting task execution. Similarly, the dependency on large pre-trained language models raises deployment challenges for real-time execution."**
> * We tested inference on 8×A100 GPUs (40GB memory each) using a batched implementation where multiple rollouts increase only memory usage, not inference time. Memory and time usage across refinement rounds for a single sample are shown below, where round 0 corresponds to the base Qwen model latency. (table will be included in the revised paper):
> | Refinement Rounds | Avg. Memory Usage per GPU (GB) | Wall Clock Time (s) |
> |-------------------|-------------------|-------------------|
> | 0 | 3.48 | 7.95 |
> | 1 | 5.84 | 9.30 |
> | 2 | 6.12 | 10.50 |
> | 3 | 6.31 | 10.95 |
> * ARMOR adds minimal computational overhead compared to the base model—approximately 1s increase in wall clock time per round, with modest memory overhead (from 3.48GB to 6.31GB per GPU). To balance performance and efficiency, one can choose to use fewer rounds of refinement (e.g. 1 of round refinement already improves reasoning by 40%)
> * ARMOR performs post-facto failure reasoning, which does not require real-time feedback. Since ARMOR is trained on execution videos, robots can complete a sequence of actions and then receive feedback from ARMOR. As tools like vLLM are still being actively developed, we believe VLM deployment optimization is an exciting area for future work to enable real-time applications.
>
> **"For the failure detection of these safety-critical tasks, using these large models with even video inputs can be overkill compared to specialized settings. Since these manipulation tasks are not performed/validated in the wild, why can one not use more overfitting metrics instead of an open-set VLM if one cares more about the binary value of success or not than the detailed reasoning? Including some discussions would further improve the soundness of the paper."**
> * While specialized discriminative models may suffice for binary failure detection in controlled settings, our core motivation is **failure reasoning**, not just detection. Failure reasoning provides actionable feedback for robot correction or human intervention, which is crucial to improve robot policy and reduce the need for expert supervision during deployment.
> * Furthermore, manipulation tasks increasingly involve diverse objects, environments, and failure modes that benefit from the generalization capabilities of VLMs. While current benchmarks may be controlled, real-world deployment will require models that can adapt to novel scenarios, which is a strength of open-set VLMs. We showcase the ability of VLMs to generalize to unseen tasks, unseen workspaces, and unseen failure modes through our transfer experiments, demonstrating their practical value beyond overfitted detectors.
> * We will add this discussion to the paper to clarify the motivation and broader impact of our approach for real-world robotic applications.

---

> > ### Author Response · Authors · 2025-11-14
> > **Response to reviewer's comments (2/2)**
> >
> > **”While the proposed multi-round self-refinement mechanism is well-motivated, the architectural backbone and training strategy primarily build upon the existing vision–language and instruction-tuning framework. The main novelty lies in integrating iterative refinement and multi-task learning, which, though valuable, represents an incremental advance rather than a new paradigm.”**
> > * We appreciate this observation and would like to clarify our contributions. First, we **modify the existing VLM architecture** by introducing separate prediction heads for different tasks while sharing the same underlying representations. This differs from prior works that use the VLM architecture as-is without task-specific adaptations, and our ablations demonstrate this multi-head design is beneficial for performance.
> > * Second, while instruction tuning is standard, the critical question lies in **how** it is done. Our results show that standard instruction-tuning algorithms (SFT-D and SFT-S+D) perform poorly on our tasks. In contrast, ARMOR employs a novel training strategy with distinct stages: offline warmup, offline expert conditioning, and online imitation learning, combined with iterative refinement at inference. This is fundamentally different from standard single-stage instruction tuning and is essential for our performance gains.
> > * Finally, we introduce a **challenging problem setting from real-world robotics**: learning effective failure reasoning under heterogeneous data availability (sparse detection labels vs. scarce reasoning annotations). This setting reflects practical constraints in robotics where obtaining detailed annotations is expensive. We demonstrate that ARMOR substantially improves upon prior works in this setting, opening avenues for future research in tackling failure reasoning under heterogeneous supervision.
> > ---
> > We believe these clarifications address the reviewer's concerns. We appreciate the thoughtful feedback and remain open to further discussion. We hope the reviewer can consider raising their score.

---

### Official Review · Reviewer_MSS9 · 2025-10-29

**Soundness:** 1
**Presentation:** 2
**Contribution:** 2
**Rating:** 2
**Confidence:** 4

**Summary:**

The paper introduces ARMOR, a VLM trained to detect failures in robotic scenarios. ARMOR introduces a two-task training pipeline separating binary failure detection and reasoning, which are iteratively optimized. A self-refinement objective further enhances reasoning ability. The authors evaluate their approach on existing failure reasoning benchmarks and show that their approach outperforms the selected baselines.

**Strengths:**

- The motivation is clear and the addressed issue is very relevant for robotics.
- The multi-task training and conditional supervision on type of available ground truth is well justified for the robotics domain.

**Weaknesses:**

### The clarity and methodology of the proposed method present several concerns

- The second stage of offline imitation seems to just reproduce the textual input. If I understand correctly, the model is provided with the ground truth labels in textual format and then learns to reproduce these outputs. This could possibly lead to the model ignoring the visual modality, making the initial training stage obsolete.
- The ablation “Multitask Prediction” also indicates this. The model almost performs on par with ARMOR, whereas reasoning is significantly worse. However, the authors don't provide the training details for this baseline, which makes it hard to tell if it's a fair comparison. The difference could be due to shorter training or absence of prompt p_t.
- It is not clear what supervision signal the model uses during online refinement. Similar methods create a refinement dataset through policy rollouts and ensure each subsequent response is better through a teacher model or reward. It seems like ARMOR solely relies on the supervision from the ground truth target reasoning without any feedback or step-by-step datasets.  This raises the question whether the iterative refinement actually results in better reasoning with multiple iterations.
- The results provided by the authors regarding performance with multiple model iterations are not convincing. Without any conditioning (step 0), the model reasons poorly, which makes sense, since the model is mainly trained on predicting reasoning conditioned on previous reasonings, and further iterations result in almost no improvements.

### The experimental section is unclear and insufficient to support the main claims
- Missing comparison with AHA, despite it being evaluated on the same benchmarks
- Including experiments showcasing the effectiveness of the method under different dense/sparse distributions (e.g., reducing the number of dense annotations further) would support the claim that the method works in regimes where data is annotated with low detail.
- Experiments showcasing reasoning transferability of reasoning from one domain to another would be more interesting to strengthen the claims of the paper (e.g. Maniskill ( M→R))
- Motivation and downstream applications (e.g., failure recovery policies, recovery reasoning…) for why correct reasoning even matters in this scenario are missing.


### Additional Issues:
- Usage of the term “two-heads” is misleading. Clarifying that the second “head” is actually just the language model head or the standard VLM would make it easier to grasp the method.
- Line 240 and subsequent:  Misleading use of “encoder-decoder” terminology for a decoder-only VLM.
- Missing model sizes
- No justification/ablation for the CLS token
- Line 322: Lowest confidence is misleading. Shouldn’t it be the lowest entropy and the highest confidence?

The paper addresses an important problem and reports promising results, but it remains unclear why the method works. The training pipeline and iterative refinement lack clear intuition and justification, and the experiments do not convincingly support the main claims. I suggest that the authors clarify the method and explain the intuition behind the iterative refinement without actually ensuring that the refinement steps result in incrementally better responses during training.

**Questions:**

- Why not use RL fine-tuning for reasoning refinement?
- How does Multitask Prediction differ from Offline Imitation?
- What model size is used, and how does performance scale?

---

> ### Author Response · Authors · 2025-11-13
> **Response to reviewer's concerns (1/3)**
>
> We thank the reviewer for their constructive feedback. We address each concern below and will incorporate clarifications in the revised paper.
>
> ---
> ### Re:  The clarity and methodology of the proposed method present several concerns
> ---
>
> First, we clarify the ablation structure of our method. The table below shows which training stages are included in each ablation. We will refer to this table in our responses below and will add it to the revised paper.
>
> | Ablation | Offline Warmup | Offline Expert Conditioning | Online Imitation | Self-Reflection (Inference) | Detection / Reasoning |
> |----------|----------------|----------------------------|------------------|----------------------------|-------|
> | Multitask Prediction | ✓ 	| 	| 	| 	| 0.897 / 0.460 |
> | Refinement Only     | ✓ 	| 	|	| ✓ 	| 0.803 / 0.488 |
> | Offline Imitation Only | ✓ 	| ✓ 	| 	| ✓ 	| 0.853 / 0.658 |
> | Online Imitation Only | 	| 	| ✓ 	| ✓ 	| 0.850 / 0.683 |
> | **ARMOR**           | ✓ 	| ✓ 	| ✓ 	| ✓	| 0.917 / 0.718 |
>
> We provide complete training details for ARMOR (number of epochs, hyperparameters) in the Appendix A.2. All ablation models are trained for the same total number of epochs as ARMOR. When an ablation excludes certain training stages, we add additional epochs to the remaining stages to ensure fair comparison. For all models, we select the best checkpoint based on validation performance on a held-out set. We will add this detail to the revised paper.
>
> **"The second stage of offline imitation seems to just reproduce the textual input. If I understand correctly, the model is provided with the ground truth labels in textual format and then learns to reproduce these outputs. This could possibly lead to the model ignoring the visual modality, making the initial training stage obsolete."**
>
> * We clarify that during offline conditional training, when providing ground-truth labels (l, e) as inputs, we **mask out one prediction** so the model learns to refine from imperfect contexts rather than overfitting. Specifically, we mask the detection input when predicting reasoning, and vice versa.  We will move this detail from the appendix to the main text for better clarity.  The "Offline Imitation Only" vs "Refinement Only" ablation demonstrates the benefit of expert conditioning.
>
> **"The ablation 'Multitask Prediction' also indicates this. The model almost performs on par with ARMOR, whereas reasoning is significantly worse. However, the authors don't provide the training details for this baseline, which makes it hard to tell if it's a fair comparison."**
>
> * Please see our training details for ablations above. For Multitask Prediction, we train the model on warm-up data for the same total epoch as ARMOR for a fair comparison. By design, Multitask Prediction lacks prompt $p_t$, which is essential for Offline Expert Conditioning, Online Imitation, and Self-Reflection. We argue that failure reasoning is a more challenging task in our setting, and the performance gap demonstrates the necessity of expert conditioning, online imitation, and self-reflection—each validated through individual ablations.
>
> **"It is not clear what supervision signal the model uses during online refinement. Similar methods create a refinement dataset through policy rollouts and ensure each subsequent response is better through a teacher model or reward. It seems like ARMOR solely relies on the supervision from the ground truth target reasoning without any feedback or step-by-step datasets.This raises the question whether the iterative refinement actually results in better reasoning with multiple iterations."**
>
> * Our online supervision signal is the ground-truth labels for detection and reasoning. We demonstrate this is sufficient: The "Offline Imitation Only" vs “ARMOR” ablation demonstrates the benefit of online refinement(+7.5% detection accuracy, +9% reasoning quality). Recursive Introspection (Qu et al. (2024)) assumes a reward model to guide refinement, which can be obtained from math problems. A verifiable reward model for failure reasoning quality can be hard to obtain, especially in our setting when the reasoning supervision is sparse. Unlike methods requiring teacher models or dense reward models, our approach works in sparse annotation regimes where additional supervision is not readily available. We leave further improvements on refinement using additional supervisions for future work.

---

> ### Author Response · Authors · 2025-11-13
> **Response to reviewer's concerns (2/3)**
>
> **"The results provided by the authors regarding performance with multiple model iterations are not convincing. Without any conditioning (step 0), the model reasons poorly, which makes sense, since the model is mainly trained on predicting reasoning conditioned on previous reasonings, and further iterations result in almost no improvements."**
>
> * Performance over refinement rounds (Figure 3(a)) in our ablation study shows consistent performance gains across rounds. We report percentage improvements at each round below. Subsequent iterations demonstrate significant gains of up to 45% in reasoning performance, showing substantial improvement for the reasoning task (which has limited training data) by leveraging the detection task (which has more training data). This highlights the importance of our approach in a heterogeneous data regime where large-scale binary labels are available but detailed reasoning labels are scarce.
> | Refinement | Round 0 | Round 1 | Round 2 | Round 3 |
> |---------|---------|---------|---------|---------|
> | Detection / Reasoning (improvements) | 0.887 / 0.488 | 0.907 / 0.683 (+2.3%, +40.0%) | 0.917 / 0.692 (+1.1%, +1.3%) | 0.917 / 0.718 (+0.0%, +3.8%) |
>
> ---
> ### Re: The experimental section is unclear and insufficient to support the main claims
> ---
>
> **"Missing comparison with AHA, despite it being evaluated on the same benchmarks."**
>
> * Our benchmark differs from AHA in the datasets used, evaluation metrics, and evaluation protocol. We improve upon several of AHA's experimental design choices, including how LLM metrics are computed (Section 5 under Metrics) and balancing of the test set (Appendix B.4 under "Evaluation Protocol"). Therefore, direct comparison with numbers reported in the AHA paper is not apple-to-apple. Given that AHA does not release their datasets or models, we adapt AHA's fine-tuning approach to evaluate its performance on our datasets. AHA uses supervised fine-tuning (SFT), and we apply AHA's training recipe using the same base model (Qwen2.5-VL) as a baseline. Both SFT-D and SFT-S+D represent two versions of the AHA baseline.
>
> **"Including experiments showcasing the effectiveness of the method under different dense/sparse distributions (e.g., reducing the number of dense annotations further) would support the claim that the method works in regimes where data is annotated with low detail."**
>
> * We provide experiments showing performance across varying ratios of sparse to dense annotations on the Sparrow→ARMBench setting. We fix the amount of sparse data from Sparrow and vary the amount of dense data in ARMBench. The column in parentheses represents the setting used in the paper.
> | Sparse / Dense Ratio | 2 | (5) | 10 | 30 |
> |----------------------|---------|---------|---------|---------|
> | Detection / Reasoning | 0.813 / 0.831 | (0.725 / 0.698) | 0.640 / 0.609 | 0.620 / 0.427 |
>
> * The results show that ARMOR is robust to up to 10× data imbalance between sparse and dense annotations (300 dense, 3000 sparse samples). However, further decreasing the dense samples causes reasoning performance to deteriorate. ARMOR scales well with increasing dense data, achieving significant performance gains when we decrease the ratio to 2 (1500 dense, 3000 sparse samples).
>
> **"Experiments showcasing reasoning transferability of reasoning from one domain to another would be more interesting to strengthen the claims of the paper (e.g. Maniskill (M→R))."**
>
> * We will explore this transfer setting if time permits. We note this is a more stringent test than our current evaluations, since no in-domain reasoning data would be available. The original motivation for our transfer setting is for the model to learn from a handful of in-domain dense data and leverage out-of-domain sparse data to improve in-domain performance.
>
> **"Motivation and downstream applications (e.g., failure recovery policies, recovery reasoning…) for why correct reasoning even matters in this scenario are missing."**
>
> * We will expand the motivation section to clarify downstream applications including failure recovery policies and recovery reasoning, following established work in this area. We will update the paper with this revision.

---

> ### Author Response · Authors · 2025-11-13
> **Response to reviewer's concerns (3/3)**
>
> ### Re: Additional Issues / Questions
> ---
>
> **"Usage of the term 'two-heads' is misleading. Clarifying that the second 'head' is actually just the language model head or the standard VLM would make it easier to grasp the method."**
>
> * We will clarify that the second "head" refers to the language model head of the standard VLM architecture in the paper.
>
> **"Line 240 and subsequent: Misleading use of 'encoder-decoder' terminology for a decoder-only VLM."**
>
> * We will correct this terminology and clarify that the Qwen VLM has a vision encoder and a language model decoder.
>
> **"Missing model sizes."**
>
> * We will add model size specifications for our method and baselines in our experiments. We finetune the 7B model of Qwen2.5VL for ARMOR and use 7B models for all the open-source baselines.
>
> **"No justification/ablation for the CLS token."**
>
> * We will cite relevant work using CLS tokens for classification and note that this approach worked well in our preliminary experiments. We leave further exploration of this design choice for future work.
>
> **"Line 322: Lowest confidence is misleading. Shouldn't it be the lowest entropy and the highest confidence?"**
>
> * We will correct "lowest confidence" to "lowest entropy/highest confidence."
>
> **"Why not use RL fine-tuning for reasoning refinement?"**
>
> * Defining a reliable reward function for failure reasoning is challenging in this domain. Our approach works in sparse annotation regimes where additional supervision is not readily available. We leave further improvements on refinement using additional supervisions for future work.
>
> **"How does Multitask Prediction differ from Offline Imitation?"**
>
> * Please see the ablation table provided above, which clarifies the exact components in each baseline. “Multitask Prediction” only does offline warmup with no expert conditioning.
>
> **"What model size is used, and how does performance scale?"**
>
> * We finetune the 7B model of Qwen2.5VL for ARMOR. We will provide complete model specifications in the paper. We will include scaling experiments with 3B and 32B variants of Qwen2.5VL if time and compute resources permit.
> ---
>
> We believe these revisions will significantly strengthen the paper's clarity on methodology and experimental support to our claims. We appreciate the reviewer's thorough feedback and hope the reviewer can consider raising their score.

---

> ### Comment · Reviewer_MSS9 · 2025-11-15
>
> Thank you for the substantial additional experiments and clarifications provided in the rebuttal. The new ablation table vastly improves the clarity of the ablations.
>
> > Specifically, we mask the detection input when predicting reasoning, and vice versa.
>
> It is still unclear to me how this would not simply result in the model reproducing its output for reasoning or failure classification, respectively. Even with the "imperfect context," i.e., a masked-out reasoning or prediction, the model could simply ignore image modalities and focus solely on the provided textual ground-truth context. How does the model perform without offline expert conditioning?
>
> >Performance over refinement rounds (Figure 3(a)) in our ablation study shows consistent performance gains across rounds.
>
> I agree that the first refinement process yields vastly improved reasoning performance. However, this makes sense since, as mentioned, the model is trained to predict reasoning conditioned on previous reasoning, thereby somewhat rendering the initial increases in performance with respect to reasoning obsolete. Additional steps show only minor improvements, raising the question if the iterative refinement is actually improving performance or if it is just some bias towards the dataset format. Given the small gains, what is the variance (hard to see from the plot)?
>
> Could the authors provide more insight into the iterative refinement process, particularly regarding the increase that occurs after the first refinement?
> How does iterative refinement help without explicit supervision? Specifically, how does the binary prediction influence reasoning accuracy during iterations?
>
> >Therefore, direct comparison with numbers reported in the AHA paper is not apple-to-apple.
>
> Thank you. While I agree, the binary success rate should be comparable. Why are the SFT-D and SFT-D+S results so much lower than in the AHA paper (0.788 vs. 1.0)? Is this due to response formatting errors? How often do such errors occur in the baseline?
>
> >Re: Additional Issues / Questions
>
> Thank you for addressing these issues. I believe they will make the method and contribution easier to comprehend.

---

> > ### Author Response · Authors · 2025-11-16
> > **Response to reviewer's further questions (1/2)**
> >
> > We thank the reviewer for their follow-up questions and would like to address them here.
> >
> > ---
> > **”It is still unclear to me how this would not simply result in the model reproducing its output for reasoning or failure classification, respectively. Even with the "imperfect context," i.e., a masked-out reasoning or prediction, the model could simply ignore image modalities and focus solely on the provided textual ground-truth context. How does the model perform without offline expert conditioning?”**
> > * We thank the reviewer for this sharp observation and would like to correct a small mistake: we mask out the ground truth **reasoning input** when predicting reasoning, not the detection input. This allows the model to leverage ground truth answers from the other task (along with the visual modality) to predict the current task. To provide valid reasoning with important details, the model cannot ignore the image modality. For instance, in Figure 3(b), despite being given the correct detection (failure: true), the model still needs to determine from the visual modality what caused the failure in the video. We admit that giving the ground truth reasoning provides higher information gain for detection. However, during online training, the model conditions on its own noisy reasoning and therefore needs to attend to the visual tokens for correct detection prediction.
> > * The "Online Imitation Only" ablation shows the model's performance without offline expert conditioning and warm-up, resulting in decreases in both detection and reasoning performance. Our experiments show no significant improvement when adding warm-up to this ablation, suggesting that expert conditioning and warm-up have a synergistic effect during offline training.

---

> ### Author Response · Authors · 2025-11-16
> **Response to reviewer's further questions (2/2)**
>
> **”I agree that the first refinement process yields vastly improved reasoning performance. However, this makes sense since, as mentioned, the model is trained to predict reasoning conditioned on previous reasoning, thereby somewhat rendering the initial increases in performance with respect to reasoning obsolete.”**
> * We do not believe that the initial improvement in reasoning is obvious or that its results are obsolete—the model is also supervised to predict correct reasoning without conditioning during warm-up, yet "Multitask Prediction" produces poor reasoning compared to ARMOR (0.460 vs. 0.718).
>
> **”Additional steps show only minor improvements, raising the question if the iterative refinement is actually improving performance or if it is just some bias towards the dataset format. Given the small gains, what is the variance (hard to see from the plot)?”**
> * The subsequent refinement steps are identical to the first step in our setting. During online training, each step uses the same input conditioning (the model's previous output) and loss supervision. Therefore, the first-step gains are not due to dataset or algorithmic bias, but rather an encouraging empirical result. We run inference with 4 seeds and report the mean and standard deviation for each refinement round below. This demonstrates that the refinements consistently improve performance with decreasing variance.
> | Refinement | Round 0 | Round 1 | Round 2 | Round 3 |
> |------------|---------|---------|---------|---------|
> | Reasoning (improvements) | 0.475±0.016  | 0.676±0.025 (+42.4%±3.5%) | 0.703±0.013 (+4.0%±2.2%) | 0.717±0.002 (+2.0%±1.0%) |
>
> **”Could the authors provide more insight into the iterative refinement process, particularly regarding the increase that occurs after the first refinement? How does iterative refinement help without explicit supervision?”**
> * In an ideal world, the model can produce the correct answer directly without any refinement. However, this is not the case in practice, as we show in our experiments and in prior works (Qu et al. 2024, Madaan et al. 2023), doing iterative inference can further improve performance. We leverage this insight to design the online training phrase, where the model is trained to output the correct predictions conditioned on imperfect inputs in each round. The online training process allows the model to attend to visual evidence when the conditionings are noisy, developing the ability to self-correct. During inference, the model sees the same distribution as online training and inherits this iterative reasoning benefit.
>
> **”Specifically, how does binary prediction influence reasoning accuracy during iterations?”**
> * In failure detection and reasoning, leveraging binary detection output can help inform the reasoning to a finer subset. In our heterogeneous data setting, detection converges faster due to greater data supervision and simpler output modality. It then guides subsequent reasoning iterations by providing accurate failure context, allowing the reasoning module to focus on explaining the correct failure mode rather than exploring incorrect hypotheses.
>
> **"Why are the SFT-D and SFT-D+S results so much lower than in the AHA paper (0.788 vs. 1.0)? Is this due to response formatting errors? How often do such errors occur in the baseline?"**
> * We thank the reviewer for pointing this out. For Maniskill results, there are two important distinctions between AHA and our setting. 1) The test dataset from AHA contains only failure examples while our test set has 50% failure and 50% success examples. A model that always predicts failure will achieve 100% accuracy in AHA. This is also why the baselines in AHA achieve as low as 0.7%, whereas a random guess should achieve 50%. We re-balance the test set to contain equal numbers of success and failure cases, which causes the performance difference. 2) The AHA model is trained on 60k data points from RLBench, whereas our model is trained on 8,000 data points from RLBench and 600 from ManiSkill. These vast differences in training data also contribute to the performance difference. The AHA (SFT) baselines have no formatting errors in our experiments. Formatting errors come from the non-finetuned baselines due to their lack of instruction-following ability, which we mitigate through prompt engineering.
>
> ---
> We hope the above responses address the reviewer's concerns and welcome any further questions.
>
> References
>
> Aman Madaan, Niket Tandon, Prakhar Gupta, Skyler Hallinan, Luyu Gao, Sarah Wiegreffe, Uri
> Alon, Nouha Dziri, Shrimai Prabhumoye, Yiming Yang, et al. Self-refine: Iterative refinement
> with self-feedback. Advances in Neural Information Processing Systems, 36:46534–46594, 2023.
>
> Yuxiao Qu, Tianjun Zhang, Naman Garg, and Aviral Kumar. Recursive introspection: Teaching
> language model agents how to self-improve. Advances in Neural Information Processing Systems,
> 37:55249–55285, 2024.

---

> ### Comment · Reviewer_MSS9 · 2025-11-17
>
> Thank you for clarifying my questions.
>  > we mask out the ground truth reasoning input when predicting reasoning, not the detection input.
>
> Thank you for clarifying. I think this is a crucial detail that should be highlighted in the main paper.
>
>  > In failure detection and reasoning, leveraging binary detection output can help inform the reasoning to a finer subset. In our heterogeneous data setting, detection converges faster due to greater data supervision and simpler output modality. It then guides subsequent reasoning iterations by providing accurate failure context, allowing the reasoning module to focus on explaining the correct failure mode rather than exploring incorrect hypotheses.
>
> Is it possible that the improved performance is simply because the model receives additional context regarding whether the rollout was successful or not? What happens if you mask the binary prediction during refinement? Does the model rely solely on this additional context and the resulting narrowed down the prediction space?
>
>
> Overall, I believe these clarifications make the method more reasonable in the applied domain. I am still not entirely convinced by the iterative refinement claims, mainly due to the low performance gains after the first refinement round. That said, the idea of performing reasoning contextualized on  high-data success-detection through joint training is interesting, and the results look promising, so I have raised my score.

---

> > ### Author Response · Authors · 2025-11-20
> > **Response to reviewer's questions**
> >
> > We sincerely thank the reviewer for the positive feedback and for raising their score! We appreciate the further insightful questions about the source of refinement improvements.
> >
> > **”Is it possible that the improved performance is simply because the model receives additional context regarding whether the rollout was successful or not?”**
> >
> > * Yes, the detection outcome provides useful insight during refinement by narrowing down the prediction space. However, we observe additional gains even after detection has converged (typically after the first refinement round), suggesting that reasoning context from previous iterations also contributes to improved performance. This is shown by the consistent improvements in rounds 2 and 3 (+6%), where detection results remain the same.
> >
> > **”What happens if you mask the binary prediction during refinement? Does the model rely solely on this additional context and the resulting narrowed down the prediction space?”**
> >
> > * If we mask out the binary detection during inference, we would expect performance to drop as the model loses the narrowed prediction space. However, the model should still improve compared to no refinement by leveraging reasoning context from previous iterations.
> >
> > * A key challenge in conducting this ablation fairly is that during online reflection training, the model is conditioned on both detection and reasoning predictions. Masking detection only at inference would create an OOD sample. To make a fair comparison, we would also need to modify the training procedure to match the inference-time masking scheme. We acknowledge this as an interesting ablation and will conduct this experiment if time permits.
> >
> > **On iterative refinement gains**
> > * We acknowledge the reviewer's observation about diminishing returns after the first round. For practical applications, a single refinement round may offer the best performance-efficiency tradeoff, and we have added this discussion to the paper in Section 5.4.

---

### Official Review · Reviewer_VQku · 2025-11-01

**Soundness:** 3
**Presentation:** 3
**Contribution:** 3
**Rating:** 6
**Confidence:** 5

**Summary:**

This paper introduces ARMOR which is used for robotic failure detection and reasoning. The authors cast binary success/failure detection and natural-language reasoning as a multi-task self-refinement process. ARMOR iteratively predicts both tasks, conditions on its prior outputs, and refines over several rounds. Training combines offline imitation (expert-conditioned supervision) and online refinement (rollouts with heterogeneous supervision: large-scale sparse binary labels + small-scale dense reasoning labels).

At inference, multiple refinement trajectories are sampled and the most confident one is selected via an entropy-based self-certainty metric. Across four datasets (RLBench-Fail, Maniskill-Fail, Sparrow-Fail, ARMBench), ARMOR reports state-of-the-art detection and reasoning performance. The work targets the important challenge of scalable, explainable robotic monitoring under limited annotations.

**Strengths:**

1. ARMOR reframes robotic failure detection and reasoning as a multi-round, multi-task self-refinement problem. Instead of predicting both tasks jointly in one forward pass, it trains the model to condition on its own past predictions (for both detection and reasoning) and iteratively improve them, which I believe is a interesting approach.

2. The proposed refinement framework is not limited to failure detection. Its iterative conditioning mechanism and multi-task training setup could naturally extend to other vision-language domains such as task planning, human-robot interaction, or embodied reasoning areas where explanation quality and temporal consistency are often more important than raw classification accuracy. This gives the method strong potential impact beyond the specific application studied.

**Weaknesses:**

1. All reported numbers appear to be single-run result, no mention of multiple seeds, error bars, or statistical significance. Given that inference involves stochastic sampling of trajectories (M=3), performance could vary substantially between runs. Without variance or confidence intervals, large reported gains (up to +100% in reasoning) cannot be verified as statistically meaningful.

2. The paper only compares to generative VLMs (Qwen2.5-VL, Cosmos-Reasoning, LLaVA-NeXT, Claude-3.7). It omits discriminative baselines (like CLIP or SigLIP fine-tuned on detection labels), which are often stronger for classification tasks.

3. The model chooses the best refinement trajectory using entropy (sec 4.3 eq 2) and entropy is used as a proxy for confidence. The paper assumes entropy correlates with correctness, but never tests this hypothesis. There’s no analysis of correlation between low entropy and high accuracy, nor justification for the λ=0.1 choice.

4. Fine-tuning Qwen2.5-VL on 8×H100 GPUs and performing multiple inference rollouts is computationally heavy. The paper does not report inference time, memory usage, or cost, which limits practical value for robotic systems requiring near-real-time feedback.

5. Although not major the ablation study in Table 2 is informative but remains limited in scope. It explores only a few coarse variants of the method and omits key hyperparameter sensitivities that are central to understanding ARMOR’s behavior. In particular, the paper does not analyze how performance varies with the number of refinement rounds (T), the number of sampled trajectories (M), or the masking probability used during offline conditional training. Similarly, the impact of balancing the detection (BCE) and reasoning (NTP) losses is not reported, which makes it difficult to gauge how each objective contributes to the final performance.

**Questions:**

1. The paper uses LLM Fuzzy Match and ROUGE-L to measure reasoning quality, but both metrics can be biased or insensitive to factual correctness. Did you conduct any human evaluation, factual consistency analysis, or use alternative metrics (e.g., BLEU, GPT-4-based scoring) to confirm that the improvements reflect genuinely better reasoning rather than surface-level similarity?

2. Since heterogeneous supervision is central to your claim, is it possible to provide evidence that the method’s improvements are not driven by cross-domain artifacts in Table 4 (e.g., using sparse data from one environment and dense data from another)? How robust is ARMOR to changes in this supervision distribution?

3. Since the detection component is a binary classification task, did you consider comparing against smaller vision-only models (e.g., ResNet, ViT, EfficientNet) fine-tuned for failure detection?

---

> ### Author Response · Authors · 2025-11-14
> **Response to reviewer's comments (1/3)**
>
> We genuinely thank the reviewer for their constructive feedback. We address each concern below and will incorporate clarifications in the revised paper.
>
> **”All reported numbers appear to be single-run result, no mention of multiple seeds, error bars, or statistical significance. Given that inference involves stochastic sampling of trajectories (M=3), performance could vary substantially between runs. Without variance or confidence intervals, large reported gains (up to +100% in reasoning) cannot be verified as statistically meaningful.”**
> * We follow AHA's evaluation protocol by fixing the same sampling parameters (which will be reported in the revision) for all models and baselines, and computing average performance over test samples. We do not tune sampling parameters for any method. We will conduct experiments using multiple seeds for inference to investigate the effect of randomness on ARMOR and report our findings.
>
> **"The paper only compares to generative VLMs (Qwen2.5-VL, Cosmos-Reasoning, LLaVA-NeXT, Claude-3.7). It omits discriminative baselines (like CLIP or SigLIP fine-tuned on detection labels), which are often stronger for classification tasks."**
> * Video classification is challenging for CLIP/SigLIP architectures. We compare ARMOR's classifier with the state-of-the-art multi-scale vision transformer (MViT) from the ARMBench paper (Mitash et al. 2023) on the ARMBench video defect dataset. ARMOR achieves comparable accuracy (0.813 vs. 0.815 for MViT). This confirms the claims by Zhang et al (2024) that generative VLMs can perform on par with discriminative VLMs when fine-tuned with classification data.
>
> * More importantly, we focus on failure reasoning to provide actionable feedback for robot correction or human intervention, but classification models cannot generate natural language reasoning, which is the reason they are not included as a baseline. While a separate classification model is possible, solving both tasks jointly allows the model to leverage shared internal representations for better reasoning while reducing computational overhead.
>
> **”The model chooses the best refinement trajectory using entropy (sec 4.3 eq 2) and entropy is used as a proxy for confidence. The paper assumes entropy correlates with correctness, but never tests this hypothesis. There’s no analysis of correlation between low entropy and high accuracy, nor justification for the λ=0.1 choice.”**
> * Figure 3(a) demonstrates the correlation between low entropy and high performance: as the number of refinement rounds increases, the output entropy decreases and performance increases. We will test this hypothesis with a statistical test and report our findings. We justify the decision for λ in our appendix: “For refinement, we set λ = 0.1 to place greater weight on detection entropy, since the detection head benefits from supervision on both D_sparse and D_dense and is thus more reliable than the explanation predictions.”
>
> **”Fine-tuning Qwen2.5-VL on 8×H100 GPUs and performing multiple inference rollouts is computationally heavy. The paper does not report inference time, memory usage, or cost, which limits practical value for robotic systems requiring near-real-time feedback.”**
> * Inference requires significantly less compute than training. We tested inference on 8×A100 GPUs (40GB memory each) using a batched implementation where multiple rollouts increase only memory usage, not inference time. Memory and time usage across refinement rounds of a single sample are shown below (table will be included in the revised paper).
> | Refinement Rounds| Avg. Memory Usage per GPU (GB) | Wall Clock Time (s) |
> |-------------------|-------------------|-------------------|
> | 0 | 3.48 | 7.95 |
> | 1 | 5.84 | 9.30 |
> | 2 | 6.12 | 10.50 |
> | 3 | 6.31 | 10.95 |
>
> * ARMOR performs post-facto failure reasoning, which does not require real-time feedback. Since ARMOR is trained on execution videos, robots can complete a sequence of actions and then receive feedback from ARMOR. Future work can explore leveraging tools for faster VLM inference as well as doing execution and reasoning in a closed-loop fashion to further improve efficiency.

---

> ### Author Response · Authors · 2025-11-14
> **Response to reviewer's comments (2/3)**
>
> **"Although not major the ablation study in Table 2 is informative but remains limited in scope. It explores only a few coarse variants of the method and omits key hyperparameter sensitivities that are central to understanding ARMOR’s behavior. In particular, the paper does not analyze how performance varies with the number of refinement rounds (T), the number of sampled trajectories (M), or the masking probability used during offline conditional training. Similarly, the impact of balancing the detection (BCE) and reasoning (NTP) losses is not reported, which makes it difficult to gauge how each objective contributes to the final performance."**
>
> * Performance over refinement rounds (Figure 3(a)) in our ablation study shows consistent performance gains across rounds. We report percentage improvements at each round below:
> | Refinement | Round 0 | Round 1 | Round 2 | Round 3 |
> |------------|---------|---------|---------|---------|
> | Detection / Reasoning (improvements) | 0.887 / 0.488 | 0.907 / 0.683 (+2.3%, +40.0%) | 0.917 / 0.692 (+1.1%, +1.3%) | 0.917 / 0.718 (+0.0%, +3.8%) |
>
> * Subsequent iterations demonstrate significant gains of up to 45% in reasoning performance, showing substantial improvement for the reasoning task (which has limited training data) by leveraging the detection task (which has more training data). We observe that performance saturates after round 3, and therefore use T = 3 for our tasks.
>
> * We varied M = 3, 6, 9 and observed similar performance. We will run additional ablations on smaller values of M and report our findings.
>
> * We clarify that during offline conditional training, when providing ground-truth labels (l, e) as inputs, we **mask out one prediction** so the model learns to refine from imperfect contexts rather than overfitting. The randomness is 0.5 for masking out $l$ v.s. $e$ for conditioning. Specifically, we mask the detection input when predicting reasoning, and vice versa. We will add this detail to the main text for better clarity.
>
> * Balance BCE + NTP: The current paradigm works well—NTP loss is only added when the minibatch (x, l, e) comes from dense data with no additional weighting. We will run additional ablations on loss balance and report our findings.
>
> * The major contributions of our method are the training stages (warmup vs. conditional, offline vs. online) and refinement (with vs. without, number of steps). We ablate each component in our ablation studies and will add the following table to the revised paper for better clarity:
> | Ablation | Offline Warmup | Offline Expert Conditioning | Online Imitation | Self-Reflection (Inference) | Detection / Reasoning performance |
> |----------|----------------|----------------------------|------------------|----------------------------|-------|
> | Multitask Prediction | ✓ 	| 	| 	| 	| 0.897 / 0.460 |
> | Refinement Only     | ✓ 	| 	|	| ✓ 	| 0.803 / 0.488 |
> | Offline Imitation Only | ✓ 	| ✓ 	| 	| ✓ 	| 0.853 / 0.658 |
> | Online Imitation Only | 	| 	| ✓ 	| ✓ 	| 0.850 / 0.683 |
> | **ARMOR**           | ✓ 	| ✓ 	| ✓ 	| ✓	| 0.917 / 0.718 |

---

> ### Author Response · Authors · 2025-11-14
> **Response to reviewer's comments (3/3)**
>
> **”The paper uses LLM Fuzzy Match and ROUGE-L to measure reasoning quality, but both metrics can be biased or insensitive to factual correctness. Did you conduct any human evaluation, factual consistency analysis, or use alternative metrics (e.g., BLEU, GPT-4-based scoring) to confirm that the improvements reflect genuinely better reasoning rather than surface-level similarity?**”
> * We employ ROUGE-L as a proxy for factual correctness, as it measures the longest common subsequence (LCS) between candidate and reference text. The ground truth reasoning contains specific words and phrases that are factually accurate, which ROUGE-L captures through this matching. Our LLM Fuzzy Match metric serves as a proxy for human evaluation by comparing semantic similarity between ground truth and generated responses without overpenalizing exact word matches. This complements ROUGE-L by capturing meaning beyond surface-level n-gram overlap. We would appreciate clarification on how GPT-4-based scoring differs from our LLM Fuzzy Match approach, as both appear to leverage language models for semantic evaluation.
>
> **”Since heterogeneous supervision is central to your claim, is it possible to provide evidence that the method’s improvements are not driven by cross-domain artifacts in Table 4 (e.g., using sparse data from one environment and dense data from another)? How robust is ARMOR to changes in this supervision distribution?”**
> * We provide experiments showing performance across varying ratios of sparse to dense annotations on the Sparrow→ARMBench setting. We fix the amount of sparse data from Sparrow and vary the amount of dense data in ARMBench. The column in parentheses represents the setting used in the paper.
> | Sparse / Dense Ratio | 2 | (5) | 10 | 30 |
> |----------------------|---------|---------|---------|---------|
> | Detection / Reasoning | 0.813 / 0.831 | (0.725 / 0.698) | 0.640 / 0.609 | 0.620 / 0.427 |
>
> * The results show that ARMOR is robust to up to 10× data imbalance between sparse and dense annotations (300 dense, 3000 sparse samples). However, further decreasing the dense samples causes reasoning performance to deteriorate. ARMOR scales well with increasing dense data, achieving significant performance gains when we decrease the ratio to 2 (1500 dense, 3000 sparse samples).
>
> **”Since the detection component is a binary classification task, did you consider comparing against smaller vision-only models (e.g., ResNet, ViT, EfficientNet) fine-tuned for failure detection?”**
> * Please see our previous response. We compare ARMOR with the state-of-the-art multi-scale vision transformer (MViT) from the ARMBench paper (Mitash et al. 2023).
>
> We hope the above responses address the reviewer's concerns and strengthen the paper's clarity and experimental support. We appreciate the reviewer's thorough feedback and welcome any further questions.
>
> ---
> References:
>
> Yuhui Zhang, Alyssa Unell, Xiaohan Wang, Dhruba Ghosh, Yuchang Su, Ludwig Schmidt, and Serena Yeung-Levy. Why are visually-grounded language models bad at image classification? Advances in Neural Information Processing Systems, 37:51727–51753, 2024b.
>
> Chaitanya Mitash, Fan Wang, Shiyang Lu, Vikedo Terhuja, Tyler Garaas, Felipe Polido, and Manikantan Nambi. Armbench: An object-centric benchmark dataset for robotic manipulation. In 2023 IEEE International Conference on Robotics and Automation (ICRA), pp. 9132–9139. IEEE, 2023.

---

> > ### Author Response · Authors · 2025-11-18
> > **Updates on results**
> >
> > We would like to update the reviewer with our experiment results and further clarifications.
> >
> > * We would like to correct a small mistake regarding our previous response on masking: we mask out the ground truth **reasoning input** when predicting reasoning, not the detection input. Concretely, for each sample we randomly select one task to train and mask out its ground truth input while retaining other inputs. This prevents the model from copying the task input and allows it to focus on cross-task and visual information.
> >
> > * **Experiment with Multiple seeds.** We run ARMOR inference with 4 seeds on RLBench and report the mean and standard deviation for each refinement round below. This demonstrates that the refinements consistently improve performance with small variance.
> > | Refinement | Round 0 | Round 1 | Round 2 | Round 3 |
> > |------------|---------|---------|---------|---------|
> > | Reasoning (improvements) | 0.475±0.016  | 0.676±0.025 (+42.4%±3.5%) | 0.703±0.013 (+4.0%±2.2%) | 0.717±0.002 (+2.0%±1.0%) |
> >
> > * **Number of Online Rollouts (M).** We varied $M \in$ {1, 2, 3, 6, 9} and observed that performance remains stable for $M \geq 3$ (0.717 ± 0.002). With smaller values of M, performance decreases with higher variance: $M=1$ achieves 0.535 ± 0.118 and $M=2$ achieves 0.636 ± 0.080. These results indicate that 3 rollouts are conducive for stable performance in practice.
> >
> > * **Balance Between BCE and NTP Loss.** We tried upweighting NTP loss to be up to 5× the BCE loss and observed faster convergence but no significant effects on performance.
> >
> > ---
> > We hope that these results help address the reviewer's concerns and strengthen the paper's clarity. We remain open to further discussion and hope the reviewer will consider these revisions favorably.

---

### Author Response · Authors · 2025-11-18
**Paper revision**

We sincerely thank all reviewers for their thoughtful feedback and constructive suggestions. We have updated the manuscript to address the reviewers' concerns and improve the paper accordingly. The changes are marked in blue, and below is a list of the changes made to the main paper and appendix:
* Added discussion on applying failure reasoning to downstream applications in the introduction
* Added discussion to motivate the usage of foundation model in the introduction
* Clarified the definition of two heads in Section 4.1
* Justified the usage of CLS token in Section 4.1
* Clarified the masking procedure in Section 4.2
* Changed "lowest confidence score" to "lowest entropy score" in Section 4.3
* Added the new ablation table with each component clearly marked
* Specifed model sizes in experiments
* Added quanitative results on performance v.s. refinement rounds with multiple seeds in Section 5.2
* Added inference cost of the model in Section 5.4
* Added broader impact of the method in the conclusion
* Added sampling hyperparameters in Appendix A.2
* Added experiments testing sparse-dense data ratio in Appendix C.1
* Added experiments testing model scaling in Appendix C.1

---

### Author Response · Authors · 2025-12-01
**Summary of Reviewer Engagement for Area Chair**

Dear Area Chair,

To reduce your overhead in reviewing our rebuttal discussions, we summarize the interaction with each reviewer below. We appreciate your time and dedication to thoroughly evaluating our work in light of the disrupted discussion period.

---
### Paper Focus
Our work addresses robot failure detection and reasoning under heterogeneous data availability (abundant sparse binary labels, scarce dense reasoning annotations). We propose ARMOR, a multi-stage training approach with iterative self-refinement that leverages vision-language models to provide natural language explanations for robot failures, enabling effective learning from mixed supervision signals.

---
### Summary of Rebuttal

**Reviewer VQku (score 6) - No engagement during rebuttal**

Concerns: Statistical significance, missing baselines, limited ablations.

Response: We provided multi-seed experiments, comparison with SOTA discriminative baselines, hyperparameter ablations,  latency analysis, and robustness experiments across data distributions.

**Reviewer MSS9 (score 2→6) - Active engagement, score increased**

Initial concerns: Model ignoring visual modality during training; unclear if iterative refinement truly improves performance.

Response: Through multiple rounds of discussion, we clarified the masking strategy (masks reasoning input, not detection, forcing visual attention), provided comprehensive ablation table delineating training stages, multi-seed validation of consistent refinement improvements, and corrected AHA baseline comparison misunderstandings. Reviewer acknowledged substantial clarity improvements and raised score to 6.

**Reviewer zLbK (score 6) - No engagement during rebuttal**

Concerns: Computational cost, VLM necessity over specialized models, limited novelty.

Response: we provided detailed latency analysis, clarified failure reasoning as core contribution requiring natural language generation, emphasized architectural modifications and novel multi-stage training strategy.

**Reviewer 2SV2 (score 6) - No engagement during rebuttal**

Concerns: Runtime latency, downstream applications, cross-embodiment transfer.

Response: we provided latency analysis, clarified ARMOR's potential for downstream tasks, and demonstrated successful cross-embodiment transfer in our original experiments in the paper.

**In summary,** we have addressed all reviewers' concerns with new experiments and clarifications, and we have incorporated the changes in the revised paper. We believe the substantive improvements, particularly the productive engagement with Reviewer 2 that led to a significant score increase, demonstrate the paper's merit for acceptance. We appreciate your time and dedication to evaluating our work in light of the disrupted discussion period.

Best regards,

The Authors

---

### Meta-Review · Area_Chair_hZss · 2026-01-10

**Summary:**

The reviewers initially raised concerns regarding the paper's methodological clarity, experimental rigor, and practical deployment overhead. Specifically, they questioned:

+ Statistical Significance: The lack of multi-seed results and error bars for the reported performance gains.

+ Methodological Validity: Whether the model might ignore visual inputs during the conditional training stage by over-relying on textual ground truth.

+ Computational Cost: The unreported inference latency and memory requirements associated with multi-round refinement.

+ Baseline Comparisons: Misunderstandings regarding the re-balanced AHA baseline results.

The authors' rebuttal successfully addressed these concerns by providing new experiments and clarifications. Key resolutions included:

+ Multi-seed Validation: New results confirmed that iterative refinements consistently improve performance with small variance.

+ Masking Strategy: Authors clarified a masking procedure that prevents the model from simply reproducing textual inputs, thereby forcing attention to visual tokens.

+ Latency Analysis: A detailed table showed that each refinement round adds approximately one second of wall-clock time with modest memory growth.

+ Baseline Alignment: Authors justified the discrepancies with prior work by explaining differences in test set balancing and training data scale.

**Reviewer Concerns:**

**Addressed Concerns**

+ Statistical Rigor: The authors provided multi-seed experiments and error bars, confirming that performance gains across refinement rounds are statistically consistent.

+ Methodological Clarity: Concerns regarding the model potentially ignoring visual modalities were resolved by clarifying a task-specific masking strategy that forces the model to attend to visual tokens.

+ Computational Transparency: Detailed latency and memory usage statistics were provided, demonstrating that multi-round refinement adds manageable overhead (approx. 1s per round).

+ Baseline Fairness: The authors clarified discrepancies in baseline comparisons, particularly justifying the re-balancing of the AHA test set for a more realistic evaluation.

+ Cross-Embodiment Transfer: Evidence of successful transfer (e.g., $R \rightarrow M$) was highlighted in the existing experimental results.

**Outstanding Concerns**

+ Incremental Novelty: Reviewer zLbK noted that while the multi-task refinement is valuable, the architectural backbone primarily builds on existing VLM frameworks, representing an incremental advance.

+ Diminishing Returns: Reviewer MSS9 remained somewhat skeptical of the iterative refinement's efficacy after the first round, as subsequent gains are relatively minor.

+ Real-time Deployment: Despite provided latency data, the dependency on large 7B+ VLMs continues to pose challenges for real-time, safety-critical robotic execution8.

**Reviewer Scores:**

Reviewer VQku (Original: 6): Likely would have maintained a 6 or increased confidence. The authors provided the requested multi-seed experiments , compared ARMOR with discriminative baselines (MVIT) , and provided a thorough latency analysis , effectively resolving the reviewer’s primary technical concerns.

Reviewer MSS9 (Original: 2, Updated: 6): Having already increased their score from 2 to 6, further movement is unlikely. While they remained skeptical of diminishing returns after the first refinement round , they explicitly acknowledged that the new masking strategy and joint training for reasoning are interesting and well-supported.

Reviewer zLbK (Original: 6): Likely would have maintained a 6. The authors successfully clarified the necessity of open-ended VLMs over specialized detectors for failure reasoning and detailed the novel training stages that distinguish ARMOR from standard instruction tuning.

Reviewer 2SV2 (Original: 6): Likely would have maintained a 6. The rebuttal provided specific evidence of cross-embodiment transfer (Panda to Sawyer) and detailed the potential for downstream task integration , directly addressing the reviewer's questions about practical utility.

---

### Decision · Program_Chairs · 2026-01-26

Accept (Poster)